# Physiological Effects of Water Salinity on Metabolism and Fatty Acid Biosynthesis in the Model Fish *Fundulus heteroclitus*

**DOI:** 10.3390/ani15172549

**Published:** 2025-08-30

**Authors:** Miguel Torres-Rodríguez, Gonzalo Martínez-Rodríguez, Leandro Rodríguez-Viera, Juan Miguel Mancera, Juan Antonio Martos-Sitcha

**Affiliations:** 1Department of Biology, Faculty of Marine and Environmental Sciences, Instituto Universitario de Investigación Marina (INMAR), Campus de Excelencia Internacional del Mar (CEI_MAR), University of Cádiz, 11510 Puerto Real, Cádiz, Spain; leandro.rodriguez@uca.es (L.R.-V.); juanmiguel.mancera@uca.es (J.M.M.); juanantonio.sitcha@uca.es (J.A.M.-S.); 2IFAPA Agua del Pino, 21450 Cartaya, Huelva, Spain; 3Instituto de Ciencias Marinas de Andalucía, Consejo Superior de Investigaciones Científicas (ICMAN-CSIC), 11519 Puerto Real, Cádiz, Spain; gonzalo.martinez@csic.es

**Keywords:** teleost, lipids, metabolism, osmoregulation, fatty acids, neural tissues

## Abstract

Salinity changes in aquatic environments pose a major challenge for fish, impacting their health, metabolism, and growth. In this study, we investigated how varying salinity levels influence the physiological and metabolic responses of *Fundulus heteroclitus*, a euryhaline fish known for its tolerance to a wide range of salinities. Over a two-month period, fish were exposed to a range of salinity conditions, and assessments were conducted on growth, biochemical parameters in blood and tissues, and the expression of fatty acids-related genes in the brain and eyes. High salinity was found to reduce growth and alter key metabolic markers, including stress hormones and lipid levels. Notably, changes in fatty acid profiles were observed in both brain and eye tissues, whereas differential expression of genes associated with fatty acid biosynthesis was evident in the eye but not in the brain. These findings suggest that organs not directly involved in osmoregulation, such as the brain and eyes, may nonetheless contribute to the adaptive response of fish to high-salinity environments, although they need a general adaptation to interact with new physiological conditions. Understanding these mechanisms enhances our knowledge of fish responses to environmental stressors and may inform future strategies in aquaculture and conservation.

## 1. Introduction

Environmental salinity is a key factor that impacts fish’s physiological and metabolic processes [1,2]. In the natural environment, water salinity can fluctuate in response to the rainfall regime, the annual thermal season, or the tidal cycle, with these fluctuations being intensified as a result of climate change [3], affecting even food production systems as aquaculture [4]. This is especially relevant in estuaries or salt marshes, where hypo- and hyper-salinity conditions occur regularly and the dynamics of the rivers are highly influenced by the weather [5,6]. Euryhaline species, in particular, can adapt to varying salinity levels, showing a strong capability for osmoregulation even under acute and prolonged osmotic stress [7,8]. While most research on salinity adaptation emphasizes osmoregulation [9,10], there is limited focus on the role of energy metabolism associated with this process [11]. Existing studies on energy metabolism primarily examine physiological responses to osmotic pressure [8,12,13]. However, little is known at the molecular level, such as changes in specific signaling pathways and gene expression regulation [9,11].

To maintain homeostasis, defined as the ability to regulate the internal milieu under osmotic stress, fish spend a high amount of energy, which in some species can account for up to 50% of their total energy budget [9,12]. Efficient energy acquisition from aquafeeds requires prior digestion to break down macronutrients into absorbable units: proteins into AAs, lipids into FAs and glycerol, and carbohydrates into monosaccharides such as glucose. The availability of these bioenergetic substrates is essential for maintaining plasma osmolality at optimal levels, typically around one-third of seawater salinity, and for achieving a specific ionic composition in body fluids, including reduced concentrations of divalent ions [14,15]. In this context, the cell membrane plays a fundamental role in regulating the osmotic balance of water and solutes, acting as a semipermeable barrier between the intracellular and extracellular environments [11,16]. In fish, lipids are not only a major source of metabolic energy but also key components of cell membranes and precursors of essential metabolites [17,18]. Thus, phospholipids and cholesterol, the main lipid components of the cell membrane, interact to form a bilayer structure that regulates membrane permeability and fluidity, thereby influencing proper cellular development and function [19].

Moreover, FAs released from lipid hydrolysis serve as important energy substrates supporting fish growth and development [20], particularly LC-PUFAs such as ARA (20:4n-6), EPA (20:5n-3), and DHA (22:6n-3). Consequently, a deeper understanding of how lipid metabolism contributes to salinity acclimation in fish may be crucial for elucidating their physiological plasticity and adaptive capacity. Recently, several studies have investigated the relationship between osmotic stress and lipid metabolism in cultured teleosts, including rainbow trout (*Oncorhynchus mykiss*) [21], spotted scat (*Scatophagus argus*) [11], and Brazilian flounder (*Paralichtys orbignyanus*) [22]. All of them highlight the key role that the liver plays as a metabolic organ in osmoregulation and salinity adaptation in fish through glycogen breakdown, subsequently increasing blood glucose levels, and finally supplying energy to the gills and other osmoregulatory organs [11,23,24]. However, little attention has been given to the relationship between salinity stress and the molecular mechanisms occurring in fish neural tissues, such as the brain and eyes. In these key organs for the cognitive system and correct fish behavior [25,26], lipids play a crucial role during optimum development and functionality, where LC-PUFAs are specifically accumulated [27,28,29]. At the cellular level, these compounds are present either in free form or as structural constituents of complex lipids such as glycerophospholipids and sphingolipids. In particular, EPA and DHA, as well as LC-PUFAs incorporated into sphingomyelin and phosphatidylcholine, are critical for brain and eye integrity and function [25,26,27,28]. Accordingly, modulating LC-PUFAs biosynthesis in these tissues through environmental factors such as salinity may be critical.

The Atlantic killifish (*Fundulus heteroclitus*), naturally present and readily accessible in the salt marshes and estuarine systems of the South-West of the Iberian Peninsula (e.g., Bay of Cádiz, Spain), is a euryhaline species capable of tolerating substantial variations in environmental salinity [30]. In these environments, which are strongly influenced by rainy seasons and evaporation processes, baseline salinity levels are slightly higher than those of seawater, approaching 40 ppt. Its remarkable ability to adapt to different salinity levels, along with its ease of handling in experimental conditions and physiological similarities to other fish species, makes it an excellent biological model for studying how teleosts adapt to changes in environmental salinity [31,32,33]. Research on this species, although primarily focused on osmoregulatory organs such as the gills, head kidney, and intestine, has significantly advanced our understanding of osmoregulatory processes in fish [34,35]. Thus, *F. heteroclitus* possesses one of the most extensively characterized genomes among estuarine teleosts [36], making it a valuable model for investigating regulatory processes such as lipid metabolism in response to environmental salinity changes, including in non-osmoregulatory organs such as the brain and eyes.

Accordingly, to better understand the molecular mechanisms underlying salinity adaptation in teleosts, particularly the relationship between salinity stress and lipid metabolism, this study evaluated the effects of different environmental salinities (2, 20, 40, and 60 ppt) on the metabolic status of *F. heteroclitus*, used as a model fish species. Growth performance, as well as dynamic changes in metabolic parameters in plasma, liver, and muscle, were assessed. Additionally, plasma osmolality, TBAs, and intestinal lipase activity were measured. The study also analyzed the influence of salinity on fatty acid composition and the expression of key genes involved in LC-FAs and VLC-FAs biosynthesis (*fads* and *elovl* families) in neural tissues, specifically the brain and eyes, which optimum levels of FAs are essential for their correct development and functionality [27,28,29,37,38].

## 2. Materials and Methods

### 2.1. Fish Maintenance and Experimental Design

Atlantic killifish, approximately 12 months old, were obtained from a stock population maintained in seawater (~37–38 ppt) at the aquaculture laboratory of the Department of Biology of the University of Cádiz (Puerto Real, Cádiz, Spain) for more than 8 months. Fish were gradually acclimated to the designed salinities over a two-week period prior to the experimental assays. Then, a total of 180 adult male fish (sexually immature) were individually weighed (4.3 ± 0.5 g) and randomly distributed in twelve 80 L fiber-glass tanks (15 fish/tank) in a closed recirculatory system provided with mechanical and biological filtration (EHEIM eXperience 250, Ref. 2424020; EHEIM, Deizisau, Germany), which allowed for maintaining the four experimental salinities. All fish were kept at 19–20 °C with constant control of pH, nitrite, nitrate, and ammonia levels, oxygen saturation, and a 12L:12D photoperiod. Animals were manually fed with commercial feed (Skretting: 57% protein, 15% lipid, and 10.5% minerals; pellet size: 0.8–1.2 mm) twice a day (10:00 and 15:00 h) until apparent visual satiation (*ad libitum*); this feeding protocol was chosen to assess the putative effects of salinity on food intake. To prevent water quality deterioration and the potential proliferation of undesirable microorganisms resulting from pellet degradation, any uneaten food was carefully removed after each feeding. Thus, *F. heteroclitus* were kept at environmental salinities of 2, 20, 40 (Ctrl), and 60 ppt in triplicate, for 62 days (from March to May 2023). The control condition (Ctrl: 40 ppt) was established in accordance with the salinity range observed in the salt ponds of the Bay of Cádiz (Spain), where the presence of these fish is common. Experimental salinities were prepared by mixing seawater and dechlorinated freshwater, sourced from the general system drinking water, to obtain 2 and 20 ppt, whereas the 40 and 60 ppt were made with seawater and natural seapond salt. Salinity is expressed in parts per thousand (ppt), equivalent to ‰. For each experimental tank, a 10% daily water renovation was performed. This assay involving fish was performed according with the guidelines for animal research set forth by the Ethics and Animal Welfare Committee of the University of Cadiz, in strict agreement with the Guidelines established by the European Union (2010/63/UE) and the Spanish legislation (RD 1201/2005 and RD 53/2013) for the use of laboratory animals (Junta de Andalucía reference number 31-05-2023-038).

### 2.2. Sampling Protocols

After the experimental trial (62 days), fish were subjected to a 24 h fasting period before the final sampling to reduce the short-term effects not only induced by recent feeding but also by the differences observed during feeding regarding to feed intake among the 4 experimental salinities/groups (see below). Then, all fish (45 fish/salinity group) were sacrificed with a lethal dose of 2-phenoxyethanol (1 mL/L^−1^) and individually weighed (WM) and measured (TL). Subsequently, 24 fish (8 fish/tank) from each experimental group were randomly sampled, as described below, to obtain biological samples for later analysis in the laboratory. Blood was collected (~180–200 µL/fish) from the caudal vessel using 1 mL heparinized syringes (25,000 units/3 mL in 0.9% NaCl) and centrifuged in 1.5 mL Eppendorf tubes at 13,000 g for 3 min at 4 °C to obtain plasma. Subsequently, fish were cervically sectioned to obtain different tissues. To obtain somatic indices, the liver and gut were removed and individually weighed. Next, biopsies of liver and white muscle (~0.5 g) were collected for metabolite analysis. The whole intestine, from the pyloric caeca to the rectum, was removed, and its length was measured. Then, intestinal samples were collected to determine lipase activity. For UFAs and gene expression analysis, the brain and eye were dissected and processed individually.

All samples for metabolites (plasma, liver, and muscle), lipase (whole intestine), and FAs (brain and eye) analysis were instantly snap-frozen in liquid nitrogen and stored at −80 °C until further analysis. For gene expression analysis, the brain and eye were immediately kept in Eppendorf tubes containing 0.5 mL of RNAlater^®^ (Invitrogen™, Thermo Fisher Scientific, Walthem, MA, USA), maintained for 24 h at 4 °C, and then stored at −20 °C until further processed.

### 2.3. Growth Performance and Somatic Indices

At the end of the trial, the following growth parameters were evaluated and calculated according to the equations provided below [39]:K = (100 × body mass)/fork length^3^MG = (100 × body mass increase)/initial body massSGR = (100 × (ln final body mass—ln initial body mass))/daysFE = mass gain/total feed intake

Furthermore, organosomatic indices from the liver, viscera, and intestine were estimated according to the following equations [40]:HSI = (100 × liver mass)/fish massVSI = (100 × gut mass)/fish massILI = (100 × intestine length)/fork body length

### 2.4. Plasma and Tissue Parameters

To analyze the metabolic parameters in liver and muscle, frozen samples were homogenized and processed according to Barany et al. [9]. Briefly, representative biopsies from liver and muscle were minced individually and homogenized in 7.5 vol (*w*/*v*) of ice-cool 0.6 N perchloric acid. Homogenates were neutralized with an equal volume of 1 M KHCO_3_, then centrifuged at 3500× *g* for 30 min at 4 °C. Supernatants were collected in 0.5 mL Eppendorf tubes. For triglyceride and cholesterol determination, aliquots were taken before centrifugation. All aliquots were stored at −80 °C until metabolite analysis. Biochemical parameters were evaluated in duplicate using a spectrophotometric method (PowerWave™ 340 microplate spectrophotometer, BioTek Instruments, Winooski, VT, USA), controlled by KCjunior Software v.1.4 for Microsoft^®^ Windows (BioTek Instruments, Winooski, VT, USA), according to the methodology described in Barany et al. [9].

Metabolites in plasma, liver, and muscle were analyzed using commercial kits (SpinReact SA, St. Esteve d’en Bas, Girona, Spain), adapted for their use in 96-well microplates. These biological compounds included glucose (Ref. 1001200), lactate (Ref. 1001330), cholesterol (Ref. 41021), and triglycerides (Ref. 1001311). Plasma total protein concentration was determined with the bicinchoninic acid method using the commercial BCA kit (BCA™ Protein assay kit, Pierce, Rockford, IL, USA). Plasma osmolality was evaluated in 20 µL samples using a Fiske One-Ten vapor pressure osmometer (Fiske Associates, Advanced Instruments, Norwood, MA, USA). Plasma cortisol levels were evaluated, as described in Molina-Roque et al. [41], with the commercial Cortisol Enzyme Immunoassay Kit (Arbor Assays, K003-H1W; Ann Arbor, MI, USA) according to the manufacturer’s indications. Plasma total bile acids were evaluated with the commercial TBAs Kit (SpinReact SA, St. Esteve d’en Bas, Girona, Spain; Ref. 1001030) using a serum calibrator (SpinReact SA, St. Esteve d’en Bas, Girona, Spain; TBAs CAL, Ref. 1002290) according to the manufacturer’s indications. The method described by Keppler and Decker [42] was used to determine the tissue glycogen amount, where the glucose obtained by later glycogen breakdown with amyloglucosidase (Sigma-Aldrich^®^, Madrid, Spain; Ref. A7420) was quantified with the same glucose commercial kit (SpinReact SA, St. Esteve d’en Bas, Girona, Spain; Ref. 1001200).

### 2.5. Lipase Activity

Lipase activity in the intestine was measured using Lipase Activity Assay Kit (Sigma-Aldrich^®^, Madrid, Spain; Cat. MAK046) following the manufacturer’s instructions. The crude extracts were diluted before each analysis, and the enzyme activity was measured as an initial rate. The samples were measured in triplicate. Briefly, the lipase substrate was tempered (1 min), vortexed, and aliquoted. Standards were prepared by diluting the 100 mM glycerol stock to obtain a calibration curve (0–10 nmol glycerol). Each reaction well contained 100 µL reaction mix (93 µL buffer, 2 µL peroxidase, 2 µL enzyme mix, and 3 µL lipase substrate), to which 10–50 µL of sample or control was added. Reactions were incubated at 37 °C, and the colorimetric product was read at 570 nm. One unit of enzyme activity (U) was defined as the change in absorbance per minute per mL (∆Abs min^−1^ mL^−1^). All assays were performed using a Bio-Tek PowerWave 340 Microplate spectrophotometer using Gen5™ 2.0 data analysis software.

### 2.6. Fatty Acid Analysis

Due to the small amount of brain (7.85 ± 2.07 mg) and eye (13.20 ± 3.52 mg) samples (previously freeze-dried at −80 °C for 48 h) from *F. heteroclitus*, FAME was obtained through an adapted direct transmethylation method [43] as described in Garrido et al. [44]. Subsequently, FAs composition was determined by gas chromatography (GC) using a SYNAPT G2-S QTOF, APGC mode, the chromatograph (Waters Corporation, Milford, MA, USA) using helium as carrier gas (2 mL min^−1^ constant flow) in a GC column Agilent DB-WAX 30 m, 0.32 mm, and 0.50 μm w/Smart Key (Agilent Technologies, LifeSciences, Santa Clara, CA, USA). Sample injections (1 μL) were performed on-column, and a thermal gradient from 80 °C to 240 °C was applied over a 7 min period. FAME, reported as % of total FAs, were identified with a standard FAME pattern (Supelco 37 component FAME Mix, CRM47885).

### 2.7. Gene Analysis

#### 2.7.1. RNA Extraction and cDNA Synthesis

Whole brain and eye from *F. heteroclitus* were individually processed for total RNA extraction, after RNAlater^®^ was eliminated, using lint-free laboratory wipes, employing an Ultra Turrax^®^ T25 (IKA^®^-Werke, Staufen im Breisgau, Germany) with a dispersing tool S25N-8G, and the NucleoSpin^®^ kit (Macherey Nagel, Düren, Germany). In all cases, a RNase-free DNase on-column digestion was used to eliminate genomic DNA contamination. Finally, RNA samples were stored at −80 °C. Total RNA concentrations were measured with a Qubit^®^ 2.0 fluorimeter and the Qubit™ RNA BR kit (Invitrogen, Thermo Fisher Scientific, Walthem, MA, USA), while their qualities were assessed with a Bioanalyzer 2100 and the RNA 6000 Nano kit (Agilent Technologies, LifeSciences, Santa Clara, CA, USA). Reverse transcription was performed with the qScript™ cDNA synthesis kit (Quanta BioScience, Beverly, MA, USA) using only samples revealing an RNA integrity number greater than 8.0 and at least 100 ng of total RNA. Each reaction was carried out in a volume of 20 μL, according to the manufacturer’s instructions, and it was diluted 1/10 with 10 mM Tris-HCl, 0.1 mM EDTA (pH = 8) to get a final concentration of 500 pg μL^−1^ from input total RNA.

#### 2.7.2. Real-Time PCR

Primers for real-time PCR were designed using Primer3 software (v.0.4.0) available at http://bioinfo.ut.ee/primer3/ (accessed on 11 April 2023), from the sequences available in GenBank (Table 1) taking into account the position of introns (through blast analysis against the whole genomic sequences for this species) to leave at least one intron between forward and reverse primers, thus avoiding any potential genomic DNA contamination. Two internal reference genes, actin beta (*actb*) and eukaryotic elongation factor 1 alpha (*eef1a*), were used as internal reference genes owing to their lower than 0.5 target stability M value and lower than 0.25 CVs. All reactions were performed in a CFX Connect™ Real-Time Detection System with BioRad CFX Maestro Software v2.3 (BioRad Laboratories, Hercules, CA, USA). A pool of cDNAs from all brain and eye samples was used as a calibrator to correct inter-assay and intra-assay errors.

Before sample analyses, six 1/10 serial dilutions (from 2 ng to 20 fg) of cDNA were carried out to verify amplification efficiency and to produce a calibration curve. Those pairs of primers that showed an efficiency (E) between 90 and 110%, a determination coefficient (R^2^) higher than 0.980, and a calibration curve interpolating at least three points over six, were chosen for real-time PCR reactions. A negative control with the highest input of total RNA (2 ng) without reverse transcription was used to check for genomic DNA contamination. Negative control with water was also useful to determine the existence of artefacts like primer-dimers.

Each reaction mixture contained 0.5 μL of each specific forward and reverse primer at their best concentration, 5 μL of iQ™ SYBR^®^ Green Supermix (BioRad Laboratories, Hercules, CA, USA), and 4 μL of cDNA (input from 2 ng of total RNA). Reactions were accomplished in a volume of 10 μL using Hard-Shell^®^ Low-Profile Thin-Wall 96 White-Well Skirted PCR plates (BioRad Laboratories, Hercules, CA, USA) covered with Microseal^®^ B Adhesive Seals (BioRad Laboratories, Hercules, CA, USA). PCRs were performed with an initial denaturation and polymerase activation at 95 °C for 10 min, followed by 40 cycles of denaturation in 15 s at 95 °C, annealing and extension at 60 °C for 30 s, and finishing with a melting curve from 60 to 95 °C increasing 0.5 °C every 5 s. ΔΔCT method [45], corrected for efficiencies [46] and normalized by geometric averaging of the two internal control genes [47], i.e., *actb* and *eef1a*, were adopted for relative gene quantification. Table 1 displays primer sequences for qPCR, amplicon size (bp), and GenBank accession number for genes analyzed.

### 2.8. Statistical Analysis

All data were checked for normality and homogeneity of variance using Kolmogorov–Smirnov and Levene’s tests, respectively. Outliers were identified by ROUT method at Q = 1%. Differences among environmental salinities for all given results were analyzed by one-way analysis of variance (ANOVA, *p* ≤ 0.05) followed by Tukey HSD post-hoc test. All data are shown as the mean ± SEM (Standard Error of the Mean). The statistical software package GraphPad Prism 8.0 (GraphPad Software Inc., San Diego, CA, USA) was used to perform all statistical analyses and design figures.

## 3. Results

### 3.1. Growth Parameters and Somatic Indices

No mortality was observed during the assay period across all tested salinity levels. The amount of aquafeed ingested (g/fish/day) for each experimental group was as follows: 2 ppt, 0.098; 20 ppt, 0.078; 40 ppt, 0.062; and 60 ppt, 0.052. Growth parameters and somatic indices are presented in Table 2. Environmental salinity significantly affected growth performance. Fish acclimated to low salinity levels (2 and 20 ppt) exhibited the highest values for K, MG, and SGR, with significant differences compared to the 60 ppt group. FE was lowest in fish reared at 60 ppt, differing significantly from those maintained at other environmental salinities (2, 20, and 40 ppt). Somatic index analysis revealed significant differences between the 40 and 60 ppt groups, with the former showing higher HSI and VSI. However, no significant differences in the ILI were observed among the salinity treatments.

### 3.2. Plasma and Tissue Biochemistry Results

Biochemical results for plasma, liver, and muscle are presented in Table 3. In plasma, no significant differences were observed in TBAs, total proteins, glucose, or lactate levels across the different salinity conditions. However, plasma osmolality showed a U-shaped relationship regarding the environmental salinities tested. This parameter presented the highest value at 60 ppt, with significant differences with respect to 20 and 40 ppt groups, though no differences were observed between the 2 and 60 ppt groups. Plasma cortisol levels increased at 60 ppt compared to lower salinities, with significant differences detected only between the extreme salinities (2 vs. 60 ppt). Lipid metabolites revealed a similar pattern for triglycerides and cholesterol, with their lowest plasma concentrations at 60 ppt.

No significant differences were found in hepatic glycogen, triglycerides, or cholesterol levels among salinity groups. Lactate and free glucose levels followed a similar trend, with the lowest concentrations at 2 and 20 ppt, and the highest at 40 and 60 ppt. In muscle tissue, lactate and cholesterol levels remain unaffected by salinity. However, glucose and triglycerides exhibited an inverse linear relationship with respect to environmental salinity, with the highest concentrations observed at 2 ppt and the lowest at 60 ppt. Additionally, muscle glycogen levels showed significant variation, peaking at 40 ppt.

### 3.3. Intestinal Lipase Activity

Gut lipolytic activity in *F. heteroclitus* varied significantly across the environmental salinities tested. Lipase activity increased progressively with rising salinity, reaching its highest level at 60 ppt (Figure 1).

### 3.4. Fatty Acid Composition

No significant differences were observed in the total UFAs content, including total MUFAs and PUFAs, in the brain (Table 4) and eye (Table 5) of *F. heteroclitus* acclimated to different environmental salinities (2, 20, 40, and 60 ppt). In the brain, significant differences were detected in 16:1n-7, 18:1n-9, 18:2n-6, 20:3n-3, 20:4n-6, and 20:5n-3 (EPA) levels. Except for EPA, which peaked at 40 ppt, the highest concentrations of these fatty acids were found at 60 ppt. Although not statistically significant, brain levels of 22:6n-3 (DHA) exhibited an increasing trend with rising salinity, with the lowest and highest values at 2 and 60 ppt, respectively (Table 4).

In the eye, significant differences were observed only for 16:1n-7 and 20:5n-3 (EPA). Notably, EPA levels exhibited an inverse trend to that observed in the brain, decreasing with increasing salinity and reaching the highest concentration at 2 ppt and the lowest at 60 ppt (Table 5).

### 3.5. Gene Expression

In the brain, no significant differences were observed in the expression levels of the genes analyzed (Figure 2). Similarly, in the eye, the expression of *fads2a*, *fads2c*, *elovl5*, and *elovl4b* did not vary significantly across salinity conditions (Figure 3A,C,D,F). However, *fads2b* and *elovl4a* expression levels exhibited an upward trend, with expression levels progressively increasing from the lowest at 2 ppt to the highest at 60 ppt (Figure 3B,E).

## 4. Discussion

It is known that environmental salinity influences growth and metabolism in fish [7,12,48]. *Fundulus heteroclitus* exhibits the broadest range of salinity tolerance within the genus *Fundulus* [49,50]. Nevertheless, its phenotype is characterized by an expanded osmotic plasticity, particularly toward lower salinity levels [51]. In this sense, and in accordance with previous salinity tolerance assays conducted with this fish [52,53], our growth results indicate significant differences between experimental groups, with the lowest growth and feed efficiency observed in fish kept at high salinity levels. According to the described relationship between teleost growth and environment salinity [12], the lowest growth rate observed in our 60 ppt fish group could be related with the high energy rate that these fish use in osmoregulatory process, around 20–50% of the total fish energy [12,54], and the activation of the synthesis and release of hormones involved in growth control (GH and IGF-1) in fish adapted to hypo-osmotic environment [55,56]. Thus, bioenergy would be redirected to the homeostatic maintenance of osmoregulation, a highly energy-consumptive process [57], to the detriment of fish growth [11,58,59]. Additionally, the observed differences in hepatosomatic and viscerosomatic indices suggest a potentially important role for the liver and gut, organs whose involvement in fish osmoregulation remains poorly understood [11], in supporting metabolic and physiological adaptation to adverse salinity conditions in teleosts [60,61,62].

Moreover, the higher accumulation of glucose and lactate, considered as two common biological stress indicators in fish [63], could denote the prevalence of the glucose biosynthesis by anaerobic pathways as described through the Cori cycle [64,65] or even by gluconeogenesis from non-lactate substrates [66], especially in the liver of fish kept at 40 and 60 ppt. Similar results were found in *Seriola dumerili* acclimated to high environmental salinities, where free glucose was significantly accumulated at the hepatic level [9]. Even more, although carbohydrates regulation in teleost is not well clarified [67,68,69], and the previous 24 h fasting period could influence at the metabolic level, our results suggest that glucose is an important metabolic substrate that can be mobilized during fish osmotic acclimation in response to environmental salinities changes [9,68], serving as fuel for osmoregulatory homeostasis and activation of osmoregulatory organs such as gills, kidney and intestine [62].

In fish, changes in plasma osmolality respond to osmotic stress [70]. Our results denoted that *F. heteroclitus* kept in extreme salinity water (2 and 60 ppt) showed the highest plasma osmolality values, especially those acclimated to 60 ppt, mirroring the typical U-shape response found for important transporters involved in the regulation of ion trafficking, as the Na^+^-K^+^-ATPase. Even so, our results are unusual regarding the hypo-osmotic environment, since it is known that under this condition, an increase in this parameter can be observed after an acute osmotic challenge as a consequence of an acute stress caused and the endocrine activation of the HPI axis with the final cortisol release, but not after a chronic exposure to low salinity waters, and further investigation would be necessary to fully understand these intriguing results. Thus, this hyperosmotic stress could be associated with tissue dehydration [9]. Due to the vital role that metabolic water plays in maintaining physiological functions and the improved growth rates observed, the results suggest that intermediate salinity levels, around 20 ppt, are optimal for this euryhaline fish, allowing for better water balance regulation [71,72], as also demonstrated in other teleosts [12,73,74]. In accordance, the highest cortisol levels, a hormone associated with hyperosmotic stress [70], were detected in the plasma of the 60 ppt fish group. Cortisol is considered the major stress hormone in fish [75] and plays a key role in the long-term osmo-response modulating ion transport capacity via regulation of transport protein expression and synthesis, cell proliferation, and differentiation [76,77]. Thus, the increase in plasma cortisol levels associated with salinity is a consequence of chronic osmotic stress, in which fish are subjected to hypersaline conditions and, consequently, the need for higher energy mobilization for homeostatic balance of osmoregulation, thus highly influencing growth performance parameters (see above). In this sense, high plasma cortisol levels could be associated with an increase in liver gluconeogenesis and immunosuppressive processes [75,78,79,80].

Attending to lipid metabolism, digestive lipases, as pancreatic and lipases activated by bile salts, are enzymes responsible for hydrolyzing the dietary lipids, obtaining energy for the development and growth of fish [81]. Its activity depends on intestinal pH (7 to 9 in fish) and Na^+^ and Ca^2+^ ion concentration in the intestine [82]. Moreover, bile acids complement lipases’ action as lipid emulsifier [81]. In this sense, although no significant differences were detected in plasmatic TBAs levels according to salinity water, the increase of intestinal lipase activity observed in 60 ppt fish groups could suggest a higher demand for lipid substrates in response to an increase in energy requirements for sustaining over time the osmoregulatory processes consequent from salinity stress [83]. Thus, as described in other aquatic organisms [84,85,86,87], environmental salinity can have significant effects on intestinal lipase activity in *F. heteroclitus*, enhancing lipid digestion efficiency under hyperosmotic stress conditions, although they were not invested in growth performance.

As mentioned above, LC-PUFAs obtained through lipid hydrolysis play a crucial role as energy sources for fish growth and development [20], acquiring special relevance ARA, EPA, and DHA, which are considered key molecules for the correct functionality of neural tissues, where these compounds are rapidly bioaccumulated [38,88]. These compounds can be incorporated through the diet and/or biosynthesized de novo from C18 PUFA precursors through enzymatic reactions carried out by fatty acyl desaturases (Fads), which introduce double bonds or unsaturations into PUFAs substrates, with the subsequent elongation of very long-chain fatty acid (Elovl) proteins, which extend the FAs chain, adding two carbons atoms [18,89]. In this way, although with a notable functional diversity attending to the position where double bonds are introduced by these proteins into the fatty acyl chain, i.e., Δ4, Δ5, Δ6, and Δ8 activities [90,91], virtually all Fads-like desaturases from fish are Fads2 orthologues. In *F. heteroclitus*, Δ4/Δ6 Fads2 activity has been described [92,93], denoting its capacity for DHA biosynthesis via Δ6 “Sprecher pathway” [94] and more directly by the “Δ4 pathway” [91]. However, despite of the key role that DHA play in the correct development and functionality of fish brain [25,26,88], no significant differences have been detected in the expression of *elovl5* and *fads2* genes (*a*, *b*, *c* isoforms), at least after a medium-term acclimation period, although its shorter-term effect during the first days of acclimatization cannot be ruled out. Curiously, significant differences were detected in eye *fads2b* expression, peaking in the 60 ppt fish group. This could suggest that *fads2b* responds positively to high levels of environmental salinity, activating its transcription to support the biosynthesis of LC-PUFAs and maintaining enough reserve of DHA to be incorporated in specific tissues with high requirements of PUFAs such as the retina [28,95], assuring the functionality of the visual system in a hyperosmotic environment. In addition to that, both in the brain and eyes, the predominant MUFA was the palmitoleic acid (16:1n-7), showing the highest values of this compound in fish acclimated to 60 ppt. Palmitoleic acid is commonly associated with the structural phospholipids present in neural tissues [29,96]. Thus, although FAs metabolism after short-term fasting can have been affected [97], these results could be associated with maintaining the structural integrity and functionality of neural cells in response to hyperosmotic stress. Similar results are observed for n-6 fatty acids ARA (20:4n-6) and its precursor linoleic acid (18:2n-6) in fish brain, both directly related to correct integrity and water-permeability cells [98], suggesting the regulatory role that these compounds play in salinity adaptation of *F. heteroclitus.* Moreover, our results also showed differences in the EPA content of eyes from fish acclimated to 2 and 60 ppt, showing its highest values in the 2 ppt fish group. Thus, the differences observed in the PUFAs content of several essential nutrients as linoleic acid, ARA (in brain) and EPA (in eyes), ligated to the lack of differences in the expression of genes involved in its biosynthesis, suggest that these differences could be due to the effects of osmotic stress on other metabolic pathways, and the capacity to accumulate FAs in specific tissues, as well as the energy mobilization and expenditure to maintaining homeostatic balance as a consequence of osmoregulation.

Attending to Elovl4 proteins, functional assays carried out in several fish species have demonstrated that both Elovl4 isoforms can elongate LC-FAs, endogenous precursors to VLC-FAs [18,37,38,89,99,100]. Elovl4 products, which are VLC-SFAs and VLC-PUFAs, perform vital roles in vertebrate development by ensuring the optimum formation and functionality of neural tissues, where *elovl4* genes are mainly expressed and these compounds are accumulated [101,102]. Although previous assays performed with other teleosts suggest that both isoforms can participate in VLC-PUFAs and VLC-SFAs elongation [99,100,103,104], Elovl4a seems to be more efficient than Elovl4b at elongating VLC-SFAs [89,99]. Despite the specific role of VLC-FAs in vertebrates remaining not fully elucidated, and their identification in fish is still limited [105], it is known that VLC-SFAs are mostly incorporated into sphingolipids in the central nervous system [102]. In mammals, these are essential for facilitating membrane fusion of synaptic vesicles during neurotransmission processes [106,107]. In teleosts, the retina undergoes continuous neurogenesis throughout the fish’s lifespan and has the capacity to regenerate retinal cells lost due to various types of damage [108,109]. This process requires sustained biosynthesis and tissue-specific assimilation of essential compounds, including VLC-FAs. Consistently, the expression pattern of *elovl4* observed in neural tissues of *F. heteroclitus* suggests the relationship between salinity adaptation and the *elovl4a* up-regulation in fish eyes. In accordance, this observation would be associated with a predictable up-regulation in the biosynthesis of Elovl4a proteins, which points to a potential implication of Elovl4a and its main synthesis products, i.e., endogenous VLC-SFAs, in the correct maintenance and functionality of neural systems from teleost in response to salinity stress. These findings highlight the critical role of the visual system to ensure an adequate lifestyle, including feeding and predator evasion behavior, in fish exposed to osmotic stress, where impaired ocular acuity may compromise normal development and performance [37,38,99].

## 5. Conclusions

This study highlights the complex physiological and molecular adaptations of *F. heteroclitus* to varying salinity conditions, demonstrating that lipid metabolism plays a pivotal role in osmoregulation and overall metabolic homeostasis. Our findings confirm that high salinity (60 ppt) induces significant metabolic stress, as reflected by elevated plasma osmolality and cortisol levels, reduced growth performance, and increased energy mobilization (24 h fasting could influence). These effects are accompanied by increased intestinal lipase activity, suggesting a greater reliance on lipid-based energy substrates under hyperosmotic conditions.

Despite minimal changes in the overall fatty acid profiles of neural tissues, specific variations in key PUFAs such as EPA and ARA suggest localized regulation or differential tissue accumulation linked to environmental salinity. Importantly, the up-regulation of *elovl4a* in the eyes of fish acclimated to high salinity points to a tissue-specific regulatory mechanism that supports the biosynthesis of VLC-SFAs, which may be critical for maintaining visual system function under osmotic stress.

Taken together, these results provide new insights into the physiological plasticity of euryhaline teleosts and underscore the essential role of lipid metabolism and neural-specific biosynthetic pathways in salinity acclimation. Further research is warranted to clarify the roles of VLC-FAs in neural function and to explore the potential of these metabolic markers in assessing fish adaptation to environmental stressors, as well as to know if these variations in these parameters are transient or permanent after the specimens maintained at the experimental salinities return to their original or normal salinity (close to 40 ppt).

## Figures and Tables

**Figure 1 animals-15-02549-f001:**
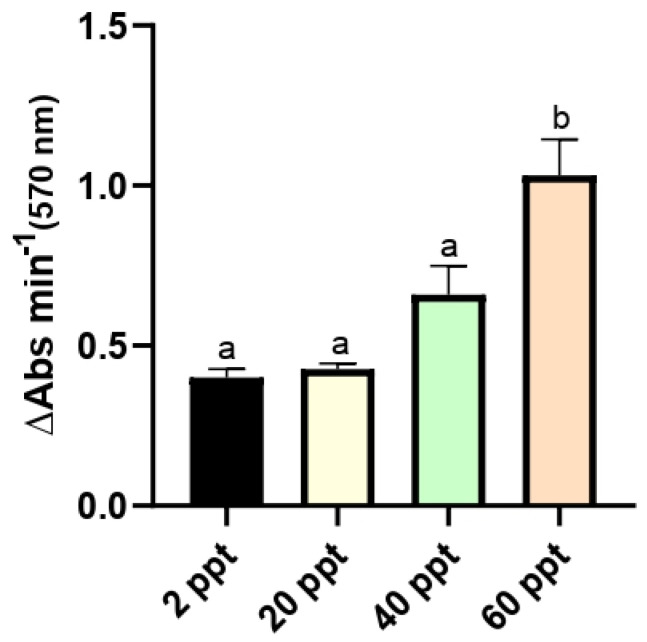
Lipase activity in the intestine of *Fundulus heteroclitus* acclimated to different environmental salinities (2, 20, 40, and 60 ppt) for 62 days. Data are presented as the mean ± SEM (n = 12). Different superscript letters (a, b) indicate significant differences among salinities based on one-way ANOVA and Tukey’s test (*p* ≤ 0.05).

**Figure 2 animals-15-02549-f002:**
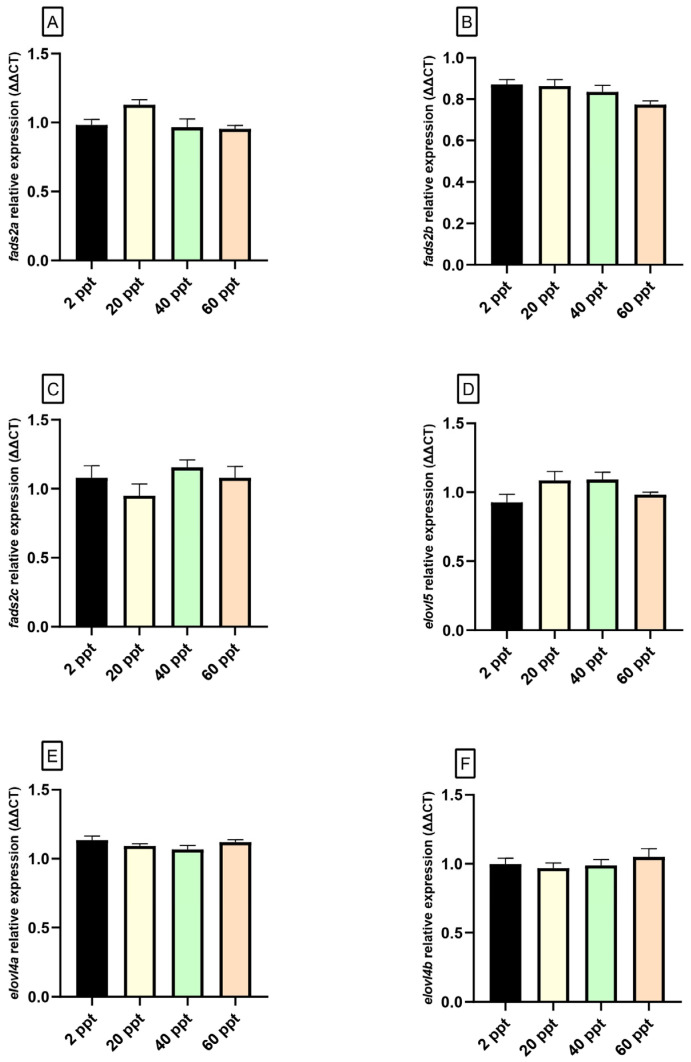
Gene expression levels (relative to *actb* and *eef1*) of brain *fads2a* (**A**), *fads2b* (**B**), *fads2c* (**C**), *elovl5* (**D**), *elovl4a* (**E**), and *elovl4b* (**F**) in *Fundulus heteroclitus* acclimated to different environmental salinities (2, 20, 40, and 60 ppt) for 62 days. Data are shown as the mean ± SEM (n = 12). No significant differences among salinities based on one-way ANOVA and Tukey’s test (*p* ≤ 0.05) were observed.

**Figure 3 animals-15-02549-f003:**
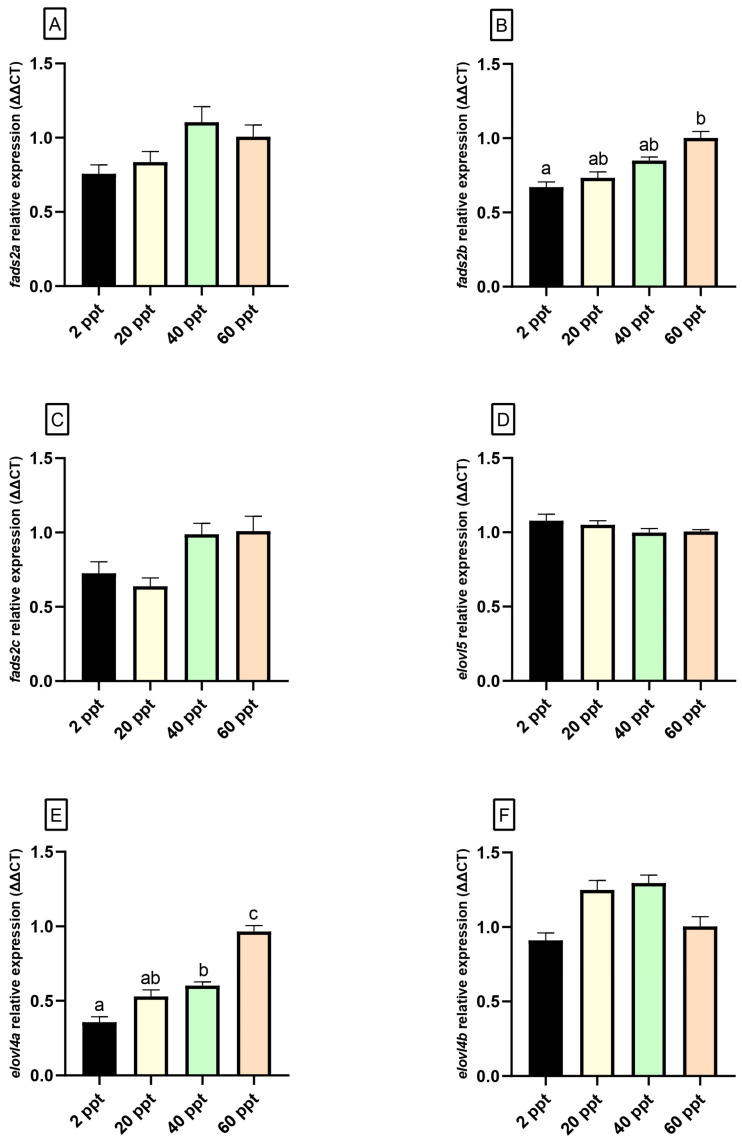
Gene expression levels (relative to *actb* and *eef1*) of eye *fads2a* (**A**), *fads2b* (**B**), *fads2c* (**C**), *elovl5* (**D**), *elovl4a* (**E**), and *elovl4b* (**F**) in *Fundulus heteroclitus* acclimated to different environmental salinities (2, 20, 40, and 60 ppt) for 62 days. Data are shown as the mean ± SEM (n = 12). Different superscript letters indicate significant differences among salinities based on one-way ANOVA and Tukey’s test (*p* ≤ 0.05).

**Table 1 animals-15-02549-t001:** Specific primers used for real-time quantitative PCR (qPCR) of *Fundulus heteroclitus* genes. Sequences of the primer pairs used (Forward: F; Reverse: R), size of the fragments produced, and accession number of the sequences used for the primer design are shown.

Gene	Genbank Acc. N	Amplicon Length (bp)	Primer Sequence (5′–3′)
*actin beta (actb)*	XM_012850364.3	163	F: GCCAACAGGGAGAAGATGAC R: CCTCGTAGATGGGCACTG
*eukaryotic elongation factor 1 alpha (eef1a)*	XM_ 012852503.3	179	F: CACCACCACAGGACACCTTA R: CAAACTTCCACAGCGAGATG
*fatty acyl desaturase 2a (fads2a)*	XM_036148888.1	200	F: CACTGGTTTGTGTGGGTGAC R: AGGTGGTAGTTGTGCCTTGG
*fatty acyl desaturase 2b (fads2b)*	XM_012865392.3	175	F: AGGACTGGCTGACCATGC R: CCGTGTTTCTCACACAGCTC
*fatty acyl desaturase 2a (fads2c)*	XM_036148890.1	183	F: AATCAAGACTGGCTGACCAT R: CACTCCGTGCTTCTCACACA
*fatty acid elongase 5 (elovl5)*	XM_036126476.1	85	F: TGTTCTCGTTCATTGTGCTTT R: TTCTGATGCTCCTTCCTTCG
*fatty acid elongase 4a (elovl4a)*	XM_012864666.3	200	F: AGGAGCCCTCTGGTGGTACT R: GGATCAGTGCGTTCATGTGT
*fatty acid elongase 4b (elovl4b)*	XM_012868850.3	177	F: TTCGGTGCAACCATCAACT R: GCAGCCAGTGTAGAGGGAAT

**Table 2 animals-15-02549-t002:** Growth performance and somatic indices of *Fundulus heteroclitus* acclimated to different environmental salinity (2, 20, 40, and 60 ppt) for 62 days.

Parameters	2 ppt	20 ppt	40 ppt	60 ppt
M_i_ (g) *	4.67 ± 0.33	3.89 ± 0.25	4.52 ± 0.39	3.82 ± 0.24
M_f_ (g) *	7.34 ± 0.34 ^b^	5.44 ± 0.20 ^a^	5.66 ± 0.26 ^a^	4.72 ± 0.15 ^a^
TL_f_ (cm) *	6.68 ± 0.11 ^b^	6.23 ± 0.11 ^a^	6.22 ± 0.11 ^a^	5.96 ± 0.08 ^a^
K *	2.47 ± 0.05 ^b^	2.44 ± 0.05 ^b^	2.38 ± 0.03 ^ab^	2.25 ± 0.03 ^a^
MG (%) *	54.20 ± 2.85 ^b^	51.54 ± 5.12 ^b^	40.16 ± 6.01 ^ab^	26.92 ± 6.33 ^a^
SGR (% day^−1^) *	0.86 ± 0.05 ^b^	0.83 ± 0.10 ^b^	0.67 ± 0.10 ^ab^	0.48 ± 0.11 ^a^
FE *	0.54 ± 0.00 ^b^	0.52 ± 0.02 ^b^	0.58 ± 0.04 ^b^	0.39 ± 0.03 ^a^
HSI (%) **	2.97 ± 0.28 ^ab^	2.68 ± 0.15 ^ab^	3.32 ± 0.21 ^b^	2.31 ± 0.21 ^a^
VSI (%) **	7.58 ± 0.39 ^ab^	7.93 ± 0.29 ^ab^	8.57 ± 0.29 ^b^	7.13 ± 0.34 ^a^
ILI (%) **	89.07 ± 4.06	93.76 ± 5.44	98.77 ± 5.39	93.55 ± 4.08

Data on growth parameters are shown as the mean ± SEM of 45 fish (*). Data on somatic indices are shown as the mean ± SEM of 24 fish (**). Different superscript letters in each row denote significant differences among water salinity fish groups (2, 20, 40, and 60 ppt) based on one-way ANOVA and Tukey test (*p* ≤ 0.05). M_i_: Initial Body Mass; M_f_: Final Body Mass; TL_f_: Final Total length; K: Fulton’s condition factor; MG: Mass Gain; SGR: Specific Growth Rate; FE: Feed Efficiency; HSI: Hepatosomatic index; VSI: Viscerosomatic index; ILI: Intestine Length Index.

**Table 3 animals-15-02549-t003:** Plasma, liver, and muscle biochemistry of *Fundulus heteroclitus* acclimated to different environmental salinities (2, 20, 40, and 60 ppt) for 62 days. Data are the mean ± SEM (n = 12). Different superscript letters indicate significant differences among salinities based on one-way ANOVA and Tukey’s test (*p* ≤ 0.05). Liver and muscle values are expressed as mmols per gram of the wet mass of tissues (g^−1^ w.m.). TBAs: Total Bile Acids.

Plasma	2 ppt	20 ppt	40 ppt	60 ppt
Osmolality (mOsm kg^−1^)	435.8 ± 19.14 ^ab^	392.7 ± 12.26 ^a^	381.4 ± 6.282 ^a^	469.8 ± 22.24 ^b^
TBAs (µmol/L)	71.94 ± 18.16	72.48 ± 13.36	67.7± 10.03	69.63 ± 14.31
Cortisol (ng/mL)	99.6 ± 19.38 ^a^	237.3 ± 34.62 ^ab^	231.4 ± 37.98 ^ab^	358.5 ± 49.21 ^b^
Proteins (mg/mL)	38.45 ± 3.26	33.97 ± 0.95	38.63 ± 0.83	31.78 ± 0.94
Glucose (mM)	5.62 ± 1.20	4.63 ± 0.41	3.91 ± 0.55	4.35 ± 0.43
Lactate (mM)	14.93 ± 1.27	14.03 ± 0.85	13.56 ± 0.65	12.71 ± 0.98
Triglycerides (mM)	5.31 ± 0.39 ^b^	3.20 ± 0.24 ^a^	4.77 ± 0.28 ^b^	2.75 ± 0.21 ^a^
Cholesterol (mM)	4.75 ± 0.16 ^b^	4.52 ± 0.21 ^ab^	5.16 ± 0.22 ^b^	3.81 ± 0.25 ^a^
**Liver**				
Glucose (mmol g^−1^ w.m.)	14.61 ± 0.91 ^a^	14.56 ± 0.67 ^a^	21.12 ± 2.05 ^b^	18.46 ± 0.23 ^b^
Glycogen (mmol g^−1^ w.m.)	43.03 ± 4.51	31.70 ± 2.07	42.21 ± 6.03	38.45 ± 4.33
Lactate (mmol g^−1^ w.m.)	7.43 ± 0.83 ^a^	5.85 ± 0.46 ^a^	12.31 ± 1.78 ^b^	11.05 ± 1.16 ^b^
Triglycerides (mmol g^−1^ w.m.)	68.42 ± 3.70	72.03 ± 5.77	67.33 ± 5.97	60.60 ± 5.52
Cholesterol (mmol g^−1^ w.m.)	27.10 ± 2.56	30.39 ± 2.53	22.69 ± 1.75	23.82 ± 2.09
**Muscle**				
Glucose (mmol g^−1^ w.m.)	2.64 ± 0.18 ^b^	2.38 ± 0.19 ^ab^	2.48 ± 0.15 ^ab^	1.88 ± 0.15 ^a^
Glycogen (mmol g^−1^ w.m.)	3.02 ± 0.50 ^a^	2.13 ± 0.43 ^a^	5.71 ± 0.82 ^b^	3.64 ± 0.63 ^ab^
Lactate (mmol g^−1^ w.m.)	21.45 ± 0.91	22.22 ± 1.94	22.29 ± 0.98	22.14 ± 1.05
Triglycerides (mmol g^−1^ w.m.)	4.48 ± 0.51 ^b^	3.22 ± 0.50 ^ab^	3.00 ± 0.44 ^ab^	2.78 ± 0.35 ^a^
Cholesterol (mmol g^−1^ w.m.)	1.79 ± 0.35	1.92 ± 0.39	1.92 ± 0.49	2.34 ± 0.53

**Table 4 animals-15-02549-t004:** Selected fatty acid content (% of total fatty acids) from the brain of *Fundulus heteroclitus* acclimated to different environmental salinities (2, 20, 40, and 60 ppt) for 62 days. Results are expressed as the mean ± SEM (n = 6). Different superscript letters denote significant differences among salinities (one-way ANOVA and Tukey test, *p* ≤ 0.05).

	Brain			
Unsaturated Fatty Acid	2 ppt	20 ppt	40 ppt	60 ppt
14:1n-5 (myristoleic acid)	0.21 ± 0.02	0.24 ± 0.03	0.23 ± 0.01	0.22 ± 0.02
16:1n-7 (palmitoleic acid)	3.51 ± 0.59 ^a^	2.89 ± 0.54 ^a^	3.08 ± 0.60 ^a^	8.08 ± 1.25 ^b^
18:1n-9 (oleic acid)	1.11 ± 0.20 ^ab^	0.92 ± 0.10 ^a^	1.34 ± 0.33 ^ab^	2.40 ± 0.57 ^b^
18:2n-6 (linoleic acid)	0.59 ± 0.10 ^ab^	0.42 ± 0.05 ^a^	0.75 ± 0.24 ^ab^	1.33 ± 0.32 ^b^
18:3n-6 (γ-linolenic acid)	0.38 ± 0.03	0.37 ± 0.05	0.35 ± 0.01	0.39 ± 0.05
18:3n-3 (α-linolenic acid)	0.30 ± 0.03	0.34 ± 0.05	0.33 ± 0.02	0.36 ± 0.04
20:3n-6 (dihomo-γ-linolenic acid)	0.37 ± 0.04	0.46 ± 0.06	0.43 ± 0.02	0.45 ± 0.05
20:3n-3 (docosatrienoic acid)	0.04 ± 0.01 ^a^	0.04 ± 0.01 ^a^	0.06 ± 0.01 ^a^	0.51 ± 0.16 ^b^
20:4n-6 (arachidonic acid)	2.45 ± 0.33 ^ab^	1.41 ± 0.26 ^a^	1.81 ± 0.25 ^a^	4.81 ± 0.85 ^b^
20:5n-3 (eicosapentaenoic acid)	3.69 ± 0.50 ^ab^	1.31 ± 0.27 ^a^	5.17 ± 0.47 ^b^	4.13 ± 0.27 ^ab^
22:1n-9 (erucic acid)	0.62 ± 0.05	0.60 ± 0.05	0.72 ± 0.13	1.06 ± 0.18
22:6n-3 (docosahexaenoic acid)	11.82 ± 2.04	12.69 ± 2.17	17.89 ± 4.06	24.12 ± 5.23
22:2n-6 (docosadienoic acid)	0.97 ± 0.15	0.76 ± 0.09	0.78 ± 0.06	0.89 ± 0.12
Total MUFAs	5.44 ± 0.86	4.64 ± 0.73	5.36 ± 1.08	11.75 ± 2.02
Total PUFAs	20.60 ± 3.22	17.8 ± 3.02	27.56 ± 5.15	37.00 ± 7.09
Total UFAs	26.05 ± 4.08	22.44 ± 3.75	32.92 ± 6.22	48.76 ± 9.11

MUFAs: Monounsaturated Fatty Acids; PUFAs: Polyunsaturated Fatty Acids; UFAs: Unsaturated Fatty Acids.

**Table 5 animals-15-02549-t005:** Selected fatty acid content (% of total fatty acids) from the eye of *Fundulus heteroclitus* acclimated to different environmental salinities (2, 20, 40, and 60 ppt) for 62 days. Results are expressed as the mean ± SEM (n = 6). Different superscript letters denote significant differences among salinities (one-way ANOVA and Tukey test, *p* ≤ 0.05).

	Eye			
Unsaturated Fatty Acid	2 ppt	20 ppt	40 ppt	60 ppt
14:1n-5 (myristoleic acid)	0.39 ± 0.04	0.24 ± 0.03	0.22 ± 0.05	0.28 ± 0.02
16:1n-7 (palmitoleic acid)	4.07 ± 0.08 ^ab^	3.83 ± 0.28 ^ab^	3.54 ± 0.24 ^a^	4.61 ± 0.10 ^b^
18:1n-9 (oleic acid)	2.74 ± 0.20	2.46 ± 0.30	2.02 ± 0.26	2.70 ± 0.09
18:2n-6 (linoleic acid)	0.93 ± 0.08	0.85 ± 0.12	0.67 ± 0.11	1.00 ± 0.03
18:3n-6 (γ-linolenic acid)	0.26 ± 0.07	0.19 ± 0.11	0.15 ± 0.11	0.21 ± 0.08
18:3n-3 (α-linolenic acid)	1.25 ± 0.16	1.27 ± 0.04	0.74 ± 0.23	1.10 ± 0.06
20:3n-6 (dihomo-γ-linolenic acid)	0.11 ± 0.01	0.12 ± 0.03	0.09 ± 0.03	0.11 ± 0.01
20:3n-3 (docosatrienoic acid)	0.08 ± 0.01	0.07 ± 0.01	0.09 ± 0.03	0.07 ± 0.01
20:4n-6 (arachidonic acid)	2.62 ± 0.22	3.14 ± 0.21	3.50 ± 0.26	2.67 ± 0.23
20:5n-3 (eicosapentaenoic acid)	1.83 ± 0.12 ^b^	1.44 ± 0.13 ^ab^	1.36 ± 0.23 ^ab^	0.84 ± 0.06 ^a^
22:1n-9 (erucic acid)	0.88 ± 0.16	0.55 ± 0.14	0.95 ± 0.39	0.80 ± 0.08
22:6n-3 (docosahexaenoic acid)	16.03 ± 0.46	19.31 ± 0.45	16.71 ± 1.68	15.12 ± 0.99
22:2n-6 (docosadienoic acid)	0.11 ± 0.01	0.13 ± 0.02	0.13 ± 0.02	0.09 ± 0.02
Total MUFAs	8.08 ± 0.48	7.08 ± 0.75	6.42 ± 0.94	8.39 ± 0.30
Total PUFAs	23.23 ± 1.11	26.52 ± 1.07	23.40 ± 2.63	21.20 ± 1.43
Total UFAs	31.31 ± 1.59	33.60 ± 1.82	29.82 ± 3.56	29.59 ± 1.73

MUFAs: Monounsaturated Fatty Acids; PUFAs: Polyunsaturated Fatty Acids; UFAs: Unsaturated Fatty Acids.

## Data Availability

Data generated in this study are available on request from the corresponding author (Miguel Torres Rodríguez; email: miguel.torres.rodriguez@juntadeandalucia.es).

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
