# Peer review of "Physiological Effects of Water Salinity on Metabolism and Fatty Acid Biosynthesis in the Model Fish *Fundulus heteroclitus"

_animals, 2025, doi:10.3390/ani15172549_

Round 1
Reviewer 1 Report
Comments and Suggestions for Authors
The study by Torres-Rodríguez et al. investigates the effects of varying salinity levels on the growth performance and metabolic biomarkers of mummichog (Fundulus heteroclitus). I find this research valuable, as salinity fluctuations in marine environments are an important consequence of climate change, with significant ecological implications. The study is, overall, well-designed and methodologically sound, offering a comprehensive dataset.
However, there are several issues in the manuscript that require revision, which I have outlined in detail below. I encourage the authors to address these concerns thoroughly and resubmit the manuscript for further evaluation.
Major issues:
- The study of the impacts of different salinities have been studied in a number of species, including mummichog. Thus, I encourage the authors to expand the discussion to clarify the novelty and specific contribution of their study to the understanding of this species’ physiology and how it may inform broader knowledge regarding salinity fluctuations and fish welfare.
- I understand the rationale for choosing a salinity gradient from hypo- to hypersalinity. Still, it would be beneficial for the authors to clarify in the introduction or discussion what these salinity levels represent ecologically (or physiologically). Given that typical seawater salinity ranges from 30–35 ppt, why was this range excluded? Including such a treatment would serve as a more ecologically relevant control group.
- I think that the description of the methods must be clarified. See for instance in Table 3 and Figure 1, the authors state n=12. Shouldn’t it be n=24?. In section 2.2: “24 fish (8 fish/tank) from each experimental group were randomly sampled, as described below, to obtain biological samples for later analysis in the laboratory”. I may have missed something. Same thing in tables 4 and 5 (n=6 ?).
- Did the authors consider including gill or kidney tissue, organs typically involved in osmoregulation, for comparison? They would seem more likely to be analysed regarding salinity fluctuations instead of muscle, for example. Also, wouldn’t it strengthen the study to have some immune-related parameters analysed in plasma and/or skin mucus? Such analyses could help understand the shifts observed by the authors.
- In the eye tissue, gene expression was upregulated, but fatty acid levels changed only modestly. Could this suggest post-transcriptional regulation?
- The authors link salinity to changes in plasma metabolites and gene expression. Can the authors clarify whether these changes may be adaptive or even stress-related? Could these responses be reversible?
- A major issue I have with this manuscript is the lack of attention to detail with abbreviations use. Several times across the manuscript, abbreviations are explained in repetition, sometimes in consecutive paragraphs. This denotes lack of attention in writing and proofreading. E.g.: line 82 repetition of the abbreviation FA; line 107 again the same abbreviation (FA); line 111 LC-PUFAs again abbreviation; section 3.4., same abbreviations; and so on. I advise the authors to carefully proofread their manuscript to avoid these mistakes
Lines 161/162 – this information belongs to the introduction section. Also, one more time the abbreviation is explained.
Minor comments:
- Need coherency in naming the species. Sometimes the authors refer to killifish, also Atlantic killifish, mummichog, then F. heteroclitus.
- Need coherency in the subsection titles. Sometimes the words begin with uppercases, sometimes with lowercases.
- Need consistency with the p value (P, P, p…). Different forms are present in the manuscript.
- Revise the graphs. The letters for statistical differences are not aligned with the bars. Also, I encourage the authors to choose a different set of colours for the bars, as also encouraged by the guidelines of Animals. This will help the reading of the paper.
Line 71: no need to nominate again glucose, AAs, lactate, and FAs; as they have been described in the sentence before.
Lines 89-90: include common names of the species.
Line 98: should be “a euryhaline,” not “an euryhaline.”
Line 127: no need to address again the scientific name.
Line 144: no need to explain the UCA abbreviation again.
Need coherency in writing the scientific units (e.g. mL L-1, fish/tank…).
Section 3.1: no need to explain again the abbreviations (SGR, FE, HIS, etc.)
Author Response
The study by Torres-Rodríguez et al. investigates the effects of varying salinity levels on the growth performance and metabolic biomarkers of mummichog (Fundulus heteroclitus). I find this research valuable, as salinity fluctuations in marine environments are an important consequence of climate change, with significant ecological implications. The study is, overall, well-designed and methodologically sound, offering a comprehensive dataset.
However, there are several issues in the manuscript that require revision, which I have outlined in detail below. I encourage the authors to address these concerns thoroughly and resubmit the manuscript for further evaluation.
Authors: We are very grateful for your suggestions and comments, and we have addressed each aspect suggested. We consider them to have greatly improved the manuscript.
Major issues:
- The study of the impacts of different salinities have been studied in a number of species, including mummichog. Thus, I encourage the authors to expand the discussion to clarify the novelty and specific contribution of their study to the understanding of this species’ physiology and how it may inform broader knowledge regarding salinity fluctuations and fish welfare.
Authors: Thank you for your suggestion. Indeed, numerous studies have investigated the impact of salinity on aquatic species, including the Atlantic killifish [1-4]. Most of these studies have mainly focused on the analysis of osmoregulatory organs, such as the gills, head kidney, and intestine. However, comparatively little attention has been devoted to the relationship between salinity stress and the molecular mechanisms occurring in non-osmoregulatory tissues, such as neural tissues present in organs like the brain and eyes. In these key organs, which are essential for cognitive processes and appropriate behavioral responses, lipids play a crucial role in ensuring optimal development and function, with LC-PUFAs being specifically bioaccumulated. Based on this, the results presented in the current manuscript highlight the critical role of the visual system in maintaining an adequate quality of life, including feeding efficiency and predator evasion, in fish exposed to osmotic stress, where impaired ocular acuity may compromise normal development and performance. Thus, this information has been highlighted in the new version of the manuscript
[1] Genz J, Grosell M. Fundulus heteroclitus acutely transferred from seawater to high salinity require few adjustments to intestinal transport associated with osmoregulation. Comp Biochem Physiol A Mol Integr Physiol 160: 156–165, 2011. doi:10.1016/j.cbpa.2011.05.027.
[2] Mancera JM, McCormick SD. 2000. Rapid activation of gill Na+,K+-ATPase in the euryhaline teleost Fundulus heteroclitus. J Exp Zool 287: 263–274.
[3] Marshall WS, Emberley TR, Singer TD, Bryson SE, McCormick SD. 1999. Time course of salinity adaptation in a strongly euryhaline estuarine teleost, Fundulus heteroclitus: a multivariable approach. J Exp Biol 202: 1535–1544.
[4] Towle DW, Gilman ME, Hempel JD. 1977. Rapid modulation of gill Na+, K+-ATPase activity during acclimation of the killfish Fundulus heteroclitus to salinity change. J Exp Zool 202: 179–186.
- I understand the rationale for choosing a salinity gradient from hypo- to hypersalinity. Still, it would be beneficial for the authors to clarify in the introduction or discussion what these salinity levels represent ecologically (or physiologically). Given that typical seawater salinity ranges from 30–35 ppt, why was this range excluded? Including such a treatment would serve as a more ecologically relevant control group.
Authors: In the southwestern coastal region of the Iberian Peninsula, these animals are commonly found in salt ponds or semi-permeable channels connected to the sea. In these environments, salinity levels are slightly higher than the typical salinity of the Atlantic Ocean (30-35 ppt), resembling more closely the salinity range observed in the Mediterranean Sea (38-40 ppt). Furthermore, salinity was assessed in the channels of various salt ponds in Puerto Real (Cádiz, Spain), where the presence of these animals is common. Based on these observations, the salinity range described in the Materials and Methods section was selected for research. In addition to that, water or salt ponds are also very influenced by seasons, where summertime can increase the salinity of these semi-closed systems even more than 40 ppt salinity. In contrast, in the rainy season, this environmental variable could greatly decrease due to the “dilution” of the system. Finally, 2 ppt was also achieved to fully understand the importance and orchestration of lipid metabolism in an environmentally real scenario where this fish species could be found. This information has also been included in the new version of our manuscript.
- I think that the description of the methods must be clarified. See for instance in Table 3 and Figure 1, the authors state n=12. Shouldn’t it be n=24?. In section 2.2: “24 fish (8 fish/tank) from each experimental group were randomly sampled, as described below, to obtain biological samples for later analysis in the laboratory”. I may have missed something. Same thing in tables 4 and 5 (n=6 ?).
Authors: Biological samples were obtained from 24 animals per group (8 fish per tank) for laboratory analysis. The extraction was performed rapidly and under cold conditions to minimize post-mortem alterations that could affect the analysis of the various parameters evaluated. However, due to the variety of techniques and tissue’s size/volume involved in the assay, the samples were subdivided to cover all intended analyses, with additional samples/aliquots preserved in case of potential degradation during handling or processing.
Based on this approach, and considering the specific characteristics of each analytical technique, including the cost per sample and the type of information provided, tissues were analysed in the number indicated in each section of the Results (n).
- Did the authors consider including gill or kidney tissue, organs typically involved in osmoregulation, for comparison? They would seem more likely to be analysed regarding salinity fluctuations instead of muscle, for example. Also, wouldn’t it strengthen the study to have some immune-related parameters analysed in plasma and/or skin mucus? Such analyses could help understand the shifts observed by the authors.
Authors: Thank you very much for your suggestion. As part of the initial experimental design, gill tissue was included for both genetic analysis and fatty acid (FA) profiling. This tissue was collected and processed alongside the other samples, and its FA profile was successfully obtained. For gene expression analysis, RNA was extracted and subsequently used for cDNA synthesis. However, during the verification of primer amplification efficiency and the generation of the calibration curve, the gill tissue did not yield satisfactory results in terms of efficiency (E), coefficient of determination (R²), and the number of points required for a reliable standard curve to perform RT-PCR reactions.
Unfortunately, for this reason and based on these results, and given the inability to associate the fatty acid profile with the expression of genes involved in its biosynthesis, it was decided not to include this tissue in the study.
- In the eye tissue, gene expression was upregulated, but fatty acid levels changed only modestly. Could this suggest post-transcriptional regulation?
Authors: Thank you for your suggestion. Indeed, various mechanisms can modify gene expression after messenger RNA (mRNA) has been transcribed but before it is translated into a protein. The gene expression analyses and tissue-specific LC-PUFA determinations carried out in our assay allow us to assess only the initial and final stages of the biosynthesis and bioaccumulation processes of these compounds, thereby overlooking a substantial portion of the metabolic cascade occurring through the fatty acid biosynthetic pathway. Moreover, subtle changes, seemingly negligible, in the concentration of essential fatty acids such as DHA can be decisive in specific organs like the eye, where these compounds not only contribute to the proper functionality of photoreceptor cells but also serve as substrates for the biosynthesis of VLC-PUFAs. These compounds, although present in very small amounts, are critical for the proper function and development of the fish neural system.
- The authors link salinity to changes in plasma metabolites and gene expression. Can the authors clarify whether these changes may be adaptive or even stress-related? Could these responses be reversible?
Authors: Thank you very much for your appreciation. Indeed, plasma metabolites mirror the energetic needs of animals undergoing any challenge, including the environmental challenge caused by changes in salinity studied in the present work. Even so, our study has not focused on the molecular regulation of these metabolites, or the synthesis of cortisol as a stress marker, which has been more extensively reviewed by other authors. Instead, it has focused on the influence of this environmental parameter, salinity, on the modifications that occur at the level of lipid metabolism in greater depth, and in relation to two very important organs of, or related to, the central nervous system, albeit with the support of the intermediary metabolism of other key players.
Furthermore, it should be noted that this saline stimulus is producing an adaptive stress to adjust the physiological processes of the specimens to the new condition. In this case, the deviations observed with respect to a salinity of 40 ppt (considered as the control, as it is quite similar to the environment where the specimens were maintained for a long period of time before the experiment) are what mark the influence of the study factor (salinity) on the analyzed parameters. In this sense, knowing whether the observed variations in these parameters are transient or permanent after the specimens are maintained at the experimental salinities and return to their original salinity (close to 40 ppt) would require new research for clarification.
- A major issue I have with this manuscript is the lack of attention to detail with abbreviations use. Several times across the manuscript, abbreviations are explained in repetition, sometimes in consecutive paragraphs. This denotes lack of attention in writing and proofreading. E.g.: line 82 repetition of the abbreviation FA; line 107 again the same abbreviation (FA); line 111 LC-PUFAs again abbreviation; section 3.4., same abbreviations; and so on. I advise the authors to carefully proofread their manuscript to avoid these mistakes
Authors: Thank you for pointing this out. We sincerely apologize for the repetition and inconsistent use of abbreviations throughout the manuscript. We understand that this reflects a lack of attention to detail and may hinder the clarity of the text. Given the high number of abbreviations used, we have not only carefully revised the manuscript to ensure that each abbreviation is defined only once at its first appearance, but we have also included a comprehensive list of abbreviations at the end of the manuscript. We believe this will help homogenize the terminology and improve the overall readability and understanding of the text. We truly appreciate your observation and have addressed it accordingly in the revised version.
Lines 161/162 – this information belongs to the introduction section. Also, one more time the abbreviation is explained.
Authors: We have removed this sentence and added it to in the introduction section as suggested.
Minor comments:
- Need coherency in naming the species. Sometimes the authors refer to killifish, also Atlantic killifish, mummichog, then F. heteroclitus.
Authors: The manuscript has been revised, and the species name has been standardized as Atlantic killifish and F. heteroclitus throughout the text to maintain consistency.
- Need coherency in the subsection titles. Sometimes the words begin with uppercases, sometimes with lowercases.
Authors: The manuscript has been revised, and the subsection titles have been standardized with lowercase to maintain consistency.
- Need consistency with the p value (P, P, p…). Different forms are present in the manuscript.
Authors: The manuscript has been revised and corrected throughout the text.
- Revise the graphs. The letters for statistical differences are not aligned with the bars. Also, I encourage the authors to choose a different set of colours for the bars, as also encouraged by the guidelines of Animals. This will help the reading of the paper.
Authors: Graphs have been revised and edited following reviewers´ suggestions.
Line 71: no need to nominate again glucose, AAs, lactate, and FAs; as they have been described in the sentence before.
Authors: Deleted.
Lines 89-90: include common names of the species.
Authors: Included.
Line 98: should be “a euryhaline,” not “an euryhaline.”
Authors: Thank you very much for the clarification. Corrected.
Line 127: no need to address again the scientific name.
Authors: Deleted.
Line 144: no need to explain the UCA abbreviation again.
Authors: Corrected.
Need coherency in writing the scientific units (e.g. mL L-1, fish/tank…).
Authors: Corrected
Section 3.1: no need to explain again the abbreviations (SGR, FE, HIS, etc.)
Authors: Deleted

Reviewer 2 Report
Comments and Suggestions for Authors
The review on the manuscript “Adaptive response to changes in environmental salinity of lipid metabolism in the model fish Fundulus heteroclitus”.
The manuscript needs a clearer justification of the study's relevance in the Introduction. The current reasoning lacks sufficient context and validation of the importance of investigating lipid metabolism adaptations in Fundulus heteroclitus. Additionally, the experimental design is inadequately described and requires a detailed rationale for the chosen trial conditions.
My review is limited to the sections prior to the Results, as I have identified several points in the manuscript that require correction and clarification.
Introduction. The authors mention that "The Atlantic killifish or mummichog (Fundulus heteroclitus) is an euryhaline species capable of tolerating significant variations in environmental salinity." However, this statement does not sufficiently justify the choice of killifish as a model organism. Species such as Gambusia or Tilapia also display similar adaptive features, and relevant studies have extensively investigated their salinity tolerance responses. The manuscript would benefit from a clear explanation of the unique characteristics or advantages of using killifish in this context.
Furthermore, the link between salinity stress and molecular mechanisms in the fish brain and eyes, as well as the proposed role of lipids, is not clearly articulated. The authors need to clarify how lipid metabolism in these tissues relates to salinity adaptation. Additionally, the significance of lipids in this process remains unclear from the current text and warrants further elaboration, what specific lipid pathways or molecules are involved, and why are they crucial?
Line 106–108: The statement that fatty acid biosynthesis enables marine fish to successfully colonize freshwater environments needs to be substantiated with mechanistic details. The manuscript would benefit, if the biological processes or pathways that facilitate this adaptation will elucidate.
On Salinity Scales. It appears that the authors have used water salinity expressed in ppt (parts per thousand) units: 2, 20, 40, and 60. Why Author chose hypersaline solution 60 ppt that is approximately 1.7 times higher than average seawater? For clarity and scientific accuracy, it is advisable to use practical salinity units (PSU), or ‰ (per mille) units.
Materials and Methods:
Line 128: It is unclear whether the fish used in this study were sourced from wild populations or aquaculture facilities. Moreover, details about the rearing conditions: were the fish kept in artificial conditions, and if so, were they transferred to laboratory settings? What was the age, sex, and reproductive status (adult or juvenile) of the fish? Since metabolism and lipid profiles can vary significantly based on these factors, this information is critical for interpretation.
Line 136: What was the control condition? You mention utilizing 18 aquaria in the trial; however, with four groups in triplicate, only 12 aquaria would be needed. Please clarify the experimental layout.
Line 139: What was the approximate amount or percentage of feed given daily to the fish? Does this amount reflect natural feeding conditions, or was it an excess designed specifically? Please specify, and provide a rationale for your feeding regime.
Line 139: Was tap water used in the study? Clarification is needed.
Line 153: What was the duration of the trial?
Experimental Design Clarification: As I understand, the design includes a 14-day acclimation period followed by XX days of exposure to different salinity conditions, ending with a 24-hour fasting period before sampling. Is this correct?
Total fish used: 180 (45 per group) for measuring total length (TL) and wet mass (WM).
Blood and tissue samples are taken from 24 fish (8 fish * 4 groups = 32 fish) and 32 fish for gene expression.
Line 153–154: A 24-hour fasting period can induce acute metabolic changes, including shifts in lipid content and fatty acid profiles, as well as other biochemical parameters. If this protocol was designed intentionally, its purpose should be explicitly justified and detailed. Fasting triggers metabolic adjustments as the fish transitions from a fed to a fasting state, influencing lipid mobilization and utilization. Notably, the fish were fed twice daily to ensure visual saturation and to simulate excess feeding conditions, which can also alter various metabolic processes. The current protocol lacks clarity regarding the precise aims related to studying lipid profiles and metabolic changes in fish under varying water salinity conditions.
Line 153: Please specify the type of syringe used for blood sampling (e.g., 1 mL) and the volume of heparin solution filled in the syringe. What was the average blood volume collected from each fish? Were all blood samples processed individually, or were they pooled per group?
Line 157: Were all tissue samples collected into tubes without any preservatives? Were all tissue and blood samples processed individually for subsequent analyses?
Section 2.3, Growth Performance and Somatic Indices. For what duration were these parameters assessed? Were these parameters measured separately at the beginning and the end of the trial? Clarification on the timing of measurements is necessary.
Line 173: The use of the K parameter is not particularly informative, as it is known to have a linear relationship with length and weight. Also, clarify whether the “3” in the text refers to a footnote or a degree notation. Please provide the minimum and maximum values for each parameter listed in Table 2.
Line 175: Why is the growth rate described as “specific”?
Line 178: What tissues do the authors refer to when mentioning “viscera”? Are gonads included? Specifically, that the formulation (viscerosomatic index) suggests that both viscera and intestines are considered.
2.4. Plasma and Tissue Parameters. Please provide details regarding the preparation of tissue samples at the beginning of this section.
Line 213. Is the statement regarding sample heating correct? Heating the lipase substrate at such high temperatures (80–100°C for 1 min) is unusual and could potentially denature the substrate or affect enzyme activity. Typically, substrates are prepared at room temperature or slightly warmed, but not boiling. Please confirm whether this step is necessary.
2.5. Fatty Acid Analysis. Please specify the volume or weight of the samples (eyes and brain), whole or tissue sections. What was the exact freeze-drying procedure?
2.6. Gene Expression. What housekeeping or reference genes were used to normalize gene expression data?
Author Response
The review on the manuscript “Adaptive response to changes in environmental salinity of lipid metabolism in the model fish Fundulus heteroclitus”.
The manuscript needs a clearer justification of the study's relevance in the Introduction. The current reasoning lacks sufficient context and validation of the importance of investigating lipid metabolism adaptations in Fundulus heteroclitus. Additionally, the experimental design is inadequately described and requires a detailed rationale for the chosen trial conditions.
My review is limited to the sections prior to the Results, as I have identified several points in the manuscript that require correction and clarification.
Authors: We are very grateful for your suggestions and comments, and we have addressed each aspect suggested. We consider them to have greatly improved the manuscript.
Introduction. The authors mention that "The Atlantic killifish or mummichog (Fundulus heteroclitus) is an euryhaline species capable of tolerating significant variations in environmental salinity." However, this statement does not sufficiently justify the choice of killifish as a model organism. Species such as Gambusia or Tilapia also display similar adaptive features, and relevant studies have extensively investigated their salinity tolerance responses. The manuscript would benefit from a clear explanation of the unique characteristics or advantages of using killifish in this context.
Authors: Thank you for your thoughtful comment. We agree that several euryhaline species, such as Gambusia and Tilapia, also exhibit notable salinity tolerance and have been widely used in osmoregulation research. However, we chose Fundulus heteroclitus (killifish) as our model organism due to several well-documented advantages that make it particularly valuable for studying environmental adaptation:
- Genomic resources: F. heteroclitus has one of the most extensively characterized genomes among estuarine teleosts, with available transcriptomic and epigenomic data that facilitate detailed molecular analyses.
- Natural populations with local adaptation: Killifish populations are well-known for their fine-scale local adaptation to environmental gradients, including salinity, temperature, and pollution, providing a unique opportunity to study physiological plasticity and evolutionary responses.
- Experimental tractability: This species is robust, easy to maintain in laboratory conditions, and has been a long-standing model in environmental physiology and toxicology, both in our research group as in others.
- Local availability: Importantly, F. heteroclitus is naturally present and easily accessible in our region, particularly in the salt marshes and estuarine systems of the Bay of Cádiz, which further facilitates its use in experimental setups and field validation.
To address your suggestion, we have revised the Introduction to highlight these attributes and better justify the selection of killifish as our model species in this study.
Furthermore, the link between salinity stress and molecular mechanisms in the fish brain and eyes, as well as the proposed role of lipids, is not clearly articulated. The authors need to clarify how lipid metabolism in these tissues relates to salinity adaptation. Additionally, the significance of lipids in this process remains unclear from the current text and warrants further elaboration, what specific lipid pathways or molecules are involved, and why are they crucial?
Authors: We appreciate the reviewer’s insightful comment. In this assay, the main lipid pathway analyzed was the biosynthesis of LC-PUFAs, assessed through quantitative analysis in neural tissues and tissue-specific expression of the genes involved in their biosynthesis (fads2, elovl5, elovl4a, elovl4b). At the cellular level, these compounds can occur either in free form or as constituents of more complex lipids, such as glycerophospholipids and sphingolipids. Thus, molecules such as sphingomyelin and phosphatidylcholine, in which LC-PUFAs form part of their structure, as well as essential fatty acids such as EPA and DHA, play a crucial role in supporting the optimal development and function of key neural organs, including the brain and eyes, where they are specifically bioaccumulated and are essential for the proper functioning of cognitive processes and the maintenance of appropriate behavioral responses. This information is included along the complete Ms, and we think that this, together with all modifications performed attending the comments stated by the Reviewers, describe the rational of the present work.
Line 106–108: The statement that fatty acid biosynthesis enables marine fish to successfully colonize freshwater environments needs to be substantiated with mechanistic details. The manuscript would benefit, if the biological processes or pathways that facilitate this adaptation will elucidate.
Authors: We appreciate the reviewer’s insightful comment. After carefully revising the Introduction, we acknowledge that the following paragraph lacked a clear connection with the specific objectives of the study:
“Thereby, F. heteroclitus can be a good model for providing knowledge about the regulation of lipid metabolism in response to changes in environmental salinity. Furthermore, the results obtained from F. heteroclitus assays underscore the potential importance of fatty acid (FAs) biosynthesis capabilities in enabling marine fish to successfully colonize freshwater environments [22, 36]. Likewise, applying this knowledge may contribute to more sustainable aquaculture practices by reducing dependence on fish oil in aquafeeds, through evaluating the capacity of certain species to modulate the biosynthesis of essential long-chain polyunsaturated fatty acids (LC-PUFAs) under varying salinity conditions [22].”
However, to improve clarity regarding the role of fatty acid biosynthesis in enabling marine fish to colonize freshwater environments successfully, we have included a brief clarification in the revised manuscript, as a response to the reviewer’s comment:
“Specifically, long-chain and very long-chain polyunsaturated fatty acids (LC-PUFAs and VLC-PUFAs) are essential for maintaining membrane fluidity and functionality under osmotic stress, particularly in epithelial tissues involved in osmoregulation. These lipids play a crucial role in regulating the activity of ion transporters and membrane-associated proteins, thereby maintaining cellular homeostasis during transitions between different salinity environments.”
We hope this paragraph could better understand the rationale intended within the work and results presented herein.
On Salinity Scales. It appears that the authors have used water salinity expressed in ppt (parts per thousand) units: 2, 20, 40, and 60. Why Author chose hypersaline solution 60 ppt that is approximately 1.7 times higher than average seawater? For clarity and scientific accuracy, it is advisable to use practical salinity units (PSU), or ‰ (per mille) units.
Authors: We thank the reviewer for this valuable comment. In this study, we used salinity values expressed in ppt (parts per thousand), which are widely used in fish physiology literature, especially when working with artificial seawater prepared by salt mixtures. While we recognize that PSU is the standard in oceanographic research, ppt is appropriate for controlled laboratory experiments where salinity is adjusted manually and not measured via conductivity. In the revised version of the manuscript, we have clarified in the Materials and Methods section that salinity values are expressed in parts per thousand (ppt), equivalent to ‰, and widely used in fish physiology and aquaculture studies. The use of ppt is consistent with the terminology adopted in many recent publications involving salinity stress in euryhaline species, including Fundulus heteroclitus.
Regarding the inclusion of a hypersaline condition (60 ppt), this decision was based on the well-documented tolerance of F. heteroclitus to a broad range of salinities, including hypersaline environments. This species naturally inhabits estuarine habitats such as tidal rivers and salt marshes, which are subject to dynamic and sometimes extreme salinity fluctuations (e.g., Tao & Breves 2024 and references therein). These fluctuations can be further exacerbated by climate change-driven factors such as increased evaporation and altered rainfall patterns. Moreover, salinity levels exceeding 58 ppt have been personally measured by one of the authors in temporary lagoons within mangrove islands along the Gulf of Mexico coastline and “esteros” in Cádiz Bay. Thus, 60 ppt represents an ecologically relevant stressor within the tolerance range of the species, allowing us to explore its physiological and molecular adaptive responses.
We hope this clarification addresses the reviewer’s concern, and we have now emphasized this rationale in the revised manuscript.
-Tao, Y. T., & Breves, J. P. (2024). Hypersalinity tolerance of mummichogs (Fundulus heteroclitus): A branchial transcriptomic analysis. Comparative Biochemistry and Physiology Part D: Genomics and Proteomics, 52, 101338.
Materials and Methods:
Line 128: It is unclear whether the fish used in this study were sourced from wild populations or aquaculture facilities. Moreover, details about the rearing conditions: were the fish kept in artificial conditions, and if so, were they transferred to laboratory settings? What was the age, sex, and reproductive status (adult or juvenile) of the fish? Since metabolism and lipid profiles can vary significantly based on these factors, this information is critical for interpretation.
Authors: We thank the reviewer for this valuable comment. We have revised the text accordingly to clarify the requested information. In this study, we selected adult male of Atlantic killifish, approximately 12 months old. These individuals were obtained from a stock population maintained in seawater (~37-38 ppt) at the aquaculture laboratory from the Department of Biology of the University of Cádiz (Puerto Real, Cádiz, Spain) for more than 8 months.
Line 136: What was the control condition? You mention utilizing 18 aquaria in the trial; however, with four groups in triplicate, only 12 aquaria would be needed. Please clarify the experimental layout.
Authors: In accordance with the salinity range observed in the channels of several salt ponds in the Bay of Cádiz, where the presence of these fish is common, the 40 ppt salinity group was used as the control condition.
As described in Section 2.1, twelve aquaria were used in this assay. “Then, a total of 180 fish were individually weighed (4.3 ± 0.5 g) and randomly distributed in twelve 80 L fiber-glass tanks…”
Line 139: What was the approximate amount or percentage of feed given daily to the fish? Does this amount reflect natural feeding conditions, or was it an excess designed specifically? Please specify, and provide a rationale for your feeding regime.
Authors: As described in Section 2.1, animals were fed twice daily (10:00 and 15:00 h) with a commercial diet (Skretting; 57% protein, 15% lipid, 10.5% minerals) until apparent visual satiation (ad libitum). This feeding strategy is commonly employed in fish trials conducted under controlled management conditions, aiming i) to balance adequate nutrition regarding the feed intake/appetite of fish depending on the experimental conditions, ii) together with optimal maintenance of water quality parameters. Feed allocation was determined by the facility technician, who adjusted the amount offered according to the observed feeding behavior of the fish in each tank, and also with the previous experience acquired during the long-time acclimation (more than 8 months) described above. Feeding was discontinued once the fish appeared satiated, allowing uneaten pellets to settle at the bottom. Consequently, the daily feed ration was not fixed but varied according to the animals’ feeding activity. The quantity of feed supplied to each experimental group was recorded weekly, enabling calculation of the average daily feed intake per fish. For clarity, feed intake was expressed as: total feed provided to each experimental group during the experimental period / number of fish / number of trial days. Following your suggestions, we have added in section 3.2 “the daily amount of aquafeed ingested (g/fish) for each experimental group was as follows: 2 ppt, 0.098; 20 ppt, 0.078; 40 ppt, 0.062; and 60 ppt, 0.052.”
Line 139: Was tap water used in the study? Clarification is needed.
Authors: Indeed, the freshwater used in the experiment was obtained from the general municipal drinking water supply. Prior to use, it was dechlorinated to ensure its suitability for the assay. This clarification has been incorporated into Section 2.1 of the revised manuscript.
Line 153: What was the duration of the trial?
Authors: As stated in Section 2.1, the experimental trial lasted 62 days. Before the trial, the animals were gradually acclimated to the corresponding salinity conditions for a period of 14 days.
Experimental Design Clarification: As I understand, the design includes a 14-day acclimation period followed by 62 days of exposure to different salinity conditions, ending with a 24-hour fasting period before sampling. Is this correct?
Authors: Yes.
Total fish used: 180 (45 per group) for measuring total length (TL) and wet mass (WM).
Authors: Effectively.
Blood and tissue samples are taken from 24 fish (8 fish * 4 groups = 32 fish) and 32 fish for gene expression.
Authors: As described in Section 2.2, biological samples were obtained from 24 animals per salinity group (8 fish per tank) for laboratory analysis. The extraction was performed rapidly and under cold conditions to minimize post-mortem alterations that could affect the analysis of the various parameters evaluated. However, due to the variety of techniques and tissues’ size/volume involved in the assay, the samples were subdivided to cover all intended analyses, with additional samples/aliquots preserved in case of potential degradation during handling or processing.
Based on this approach, and considering the specific characteristics of each analytical technique, including the cost per sample and the type of information provided, tissues were analysed in the number indicated in each section of the Results (n).
Line 153–154: A 24-hour fasting period can induce acute metabolic changes, including shifts in lipid content and fatty acid profiles, as well as other biochemical parameters. If this protocol was designed intentionally, its purpose should be explicitly justified and detailed. Fasting triggers metabolic adjustments as the fish transitions from a fed to a fasting state, influencing lipid mobilization and utilization. Notably, the fish were fed twice daily to ensure visual saturation and to simulate excess feeding conditions, which can also alter various metabolic processes. The current protocol lacks clarity regarding the precise aims related to studying lipid profiles and metabolic changes in fish under varying water salinity conditions.
Authors: Thank you for your valuable observation. We agree that fasting can trigger relevant metabolic adjustments. In our study, the 24-hour fasting period prior to sampling was intentionally implemented to reduce the short-term effects not only induced by recent feeding but also by the differences observed during feeding regarding to feed intake among the 4 experimental salinities/groups, allowing us to maintain a more standardized assessment of the basal physiological and metabolic status of the fish across salinity treatments.
Thus, this approach aimed to minimize variability in digestive and postprandial metabolic processes, which could otherwise obstruse salinity-induced effects on lipid metabolism and other biochemical parameters. Furthermore, 24-h of fasting before sampling is a common practice in metabolic and nutritional studies involving fish, especially when evaluating lipid mobilization, enzymatic activity, or gene expression related to energy balance if differences in feed intake were observed, or when dietary management could interfere with metabolic determinations by masking the main factor studied.
The feeding regimen, two meals per day to apparent satiety, was designed to ensure optimal growth, but the fasting period prior to tissue collection allowed us to separate the effects of salinity from transient responses to recent feeding.
We have now clarified this rationale in the Materials and Methods section of the revised manuscript to enhance the transparency of our experimental design.
Line 153: Please specify the type of syringe used for blood sampling (e.g., 1 mL) and the volume of heparin solution filled in the syringe. What was the average blood volume collected from each fish? Were all blood samples processed individually, or were they pooled per group?
Authors: The syringes (1 mL) were heparinized by repeatedly aspirating and expelling the following solution (25,000 units/3 mL in 0.9% NaCl), ensuring thorough contact of the internal surfaces with the anticoagulant. During this process, a minimal volume (~5 µL) of the heparin solution was retained in each syringe.
Exact values for the total volume of blood collected from each fish were not fully recorded, as the aim during sampling was to extract the maximum possible amount to obtain as much as plasma possible for all determinations required. On average, approximately 180-200 µL of blood was obtained per individual. Blood samples were processed individually.
Line 157: Were all tissue samples collected into tubes without any preservatives? Were all tissue and blood samples processed individually for subsequent analyses?
Authors: As described in section 2.2, “for gene expression analysis, the brain and eye were immediately kept in Eppendorf tubes containing 0.5 mL of RNAlater® (Invitrogen™, Thermo Fisher Scientific), maintained for 24 h at 4 °C, and then stored at -20 °C until further processed”.
Effectively, all tissues and blood samples were processed individually for analyses.
Section 2.3, Growth Performance and Somatic Indices. For what duration were these parameters assessed? Were these parameters measured separately at the beginning and the end of the trial? Clarification on the timing of measurements is necessary.
Authors: These parameters were evaluated at the end of the trial. To clarify this point, we have included the following sentence in Section 2.3: “At the end of the trial, the following growth parameters were evaluated and calculated according to the equations provided below.”
Line 173: The use of the K parameter is not particularly informative, as it is known to have a linear relationship with length and weight. Also, clarify whether the “3” in the text refers to a footnote or a degree notation. Please provide the minimum and maximum values for each parameter listed in Table 2.
Authors: Fulton’s condition factor (K) is a commonly used measure in fish studies to assess general health status, body condition, and physiological well-being. This easily measurable parameter, when evaluated alongside other zootechnical variables, provides valuable information on the body condition of fish maintained under different treatments or environmental conditions, including salinity [1, 2]. In this formula, the superscript 3 denotes that the length is raised to the power of three (i.e., cubed).
All parameters listed in Table 2, as well as those in the remaining tables and figures, are presented as mean ± SEM to allow readers to interpret the variability of the measurements clearly and intuitively. As supported by the literature published in Animals [1] and other scientific journals [3], this is a widely accepted and valid method of data presentation. Therefore, we believe that including minimum and maximum values would not enhance the interpretation of the results and might unnecessarily complicate the table conformation, presentation and the subsequent understanding by the reader.
[1] Fernandez-López, E., Panzera, Y., Bessonart, M., Marandino, A., Féola, F., Gadea, J., ... & Salhi, M. (2024). Effect of salinity on fads2 and elovl gene expression and fatty acid profile of the euryhaline flatfish Paralichthys orbignyanus. Aquaculture, 583, 740585.
[2] Overton, J. L., Bayley, M., Paulsen, H., & Wang, T. (2008). Salinity tolerance of cultured Eurasian perch, Perca fluviatilis L.: effects on growth and on survival as a function of temperature. Aquaculture, 277(3-4), 282-286.
[3] Molina-Roque, L., Simó-Mirabet, P., Barany, A., Caderno, A., Navarro-Guillén, C., Galafat, A., ... & Martos-Sitcha, J. A. (2025). Enzymatic treatment of plant proteins in combination with algae-based nutraceutical inclusion in aquafeeds improves growth performance and physiological traits in the greater amberjack (Seriola dumerili). Aquaculture, 598, 742012.
Line 175: Why is the growth rate described as “specific”?
Authors: The term "specific" in "specific growth rate" (SGR) refers to the fact that the growth rate is normalized relative to the size of the organism. Specifically, SGR expresses the percentage increase in body mass per unit of time (usually per day), accounting for the logarithmic nature of biological growth. This normalization allows for comparisons of growth across individuals or groups with different initial sizes and is a standard parameter in aquaculture and fish physiology studies. We have clarified this in the revised manuscript, if needed.
Line 178: What tissues do the authors refer to when mentioning “viscera”? Are gonads included? Specifically, that the formulation (viscerosomatic index) suggests that both viscera and intestines are considered.
Authors: Thank you for your observation. In our study, the term “viscera” refers to the internal organs, excluding the gonads. Specifically, for the calculation of the viscerosomatic index (VSI), we included the digestive tract, liver, and associated organs, but not the gonads. This is in line with commonly accepted definitions in fish physiology studies. We have now clarified this in the Materials and Methods section to avoid any ambiguity.
2.4. Plasma and Tissue Parameters. Please provide details regarding the preparation of tissue samples at the beginning of this section.
Authors: Done.
Line 213. Is the statement regarding sample heating correct? Heating the lipase substrate at such high temperatures (80–100°C for 1 min) is unusual and could potentially denature the substrate or affect enzyme activity. Typically, substrates are prepared at room temperature or slightly warmed, but not boiling. Please confirm whether this step is necessary.
Authors: Thank you for your observation. You are correct, heating the lipase substrate to 80–100°C was mistakenly included in the Materials and Methods section. Although some long-chain lipase substrates, such as palmitate, may require heating (sometimes up to 60°C) to facilitate solubilization, this was not the case in our study. The commercial kit used for lipase determination did not require substrate heating, and the assay was performed at room temperature following the manufacturer’s instructions. We have corrected this section accordingly and removed the heating step from the revised manuscript.
2.5. Fatty Acid Analysis. Please specify the volume or weight of the samples (eyes and brain), whole or tissue sections. What was the exact freeze-drying procedure?
Authors: Done. Brain (7.85 ± 2.07 mg) and eye (13.20 ± 3,52 mg) samples were lyophilized at -80ºC for 48 hours.
2.6. Gene Expression. What housekeeping or reference genes were used to normalize gene expression data?
Authors: This information is provided in subsection 2.6.2 Real time PCR as follows (lines 256-259): “Two internal reference genes, actin beta (actb) and eukaryotic elongation factor 1 alpha (eef1a), were used as internal reference genes owing to their lower than 0.5 target stability M value and lower than 0.25 CVs.”
Authors: Thank you for your thoughtful comments and valuable suggestions. We believe that your recommendations have significantly improved the overall quality and clarity of the manuscript.
Round 2
Reviewer 1 Report
Comments and Suggestions for Authors
Thank you for the authors for taking into consideration my comments.
Author Response
Dear Reviewer,
On behalf of all the co-authors, I would like to sincerely thank you for the careful evaluation of our manuscript entitled “Adaptive response to changes in environmental salinity of lipid metabolism in the model fish Fundulus heteroclitus”.
We are grateful for your constructive comments and suggestions, which have been very helpful in improving the quality and clarity of our work.
We greatly appreciate your time and consideration, and we thank you for your favorable evaluation.
Sincerely,
Miguel Torres
On behalf of all co-authors
Reviewer 2 Report
Comments and Suggestions for Authors
I appreciate the effort the authors have invested in revising the manuscript. However, the current version contains areas where clarity, detail, and rationale need further strengthening to meet the journal’s standards of scientific rigor and clarity. The mean and its error alone are insufficient as statistical parameters. Minimum and maximum values are critical as they provide essential insights into the range and distribution of the dataset, helping to identify any outliers. Please provide all the necessary data.
The shallow place of your study is feeding regimes and fasting protocols. The authors should justify their feeding and fasting protocols in relation to the specific objectives of their study, rather than solely referencing widely accepted practices in fish physiology literature.
- The 62-day exposure period constitutes a long-term experiment, likely allowing for the stabilization of physiological responses related to salinity adaptation. While the term “adaptation” can encompass numerous processes, in the context of your study, the observed responses seem to reflect the overall physiological state of fish inhabiting such salinity conditions, rather than rapid or short-term reactions to salinity changes. Since the experiments were conducted in a laboratory setting, the use of the term “environmental” in the manuscript title may be inappropriate. Based on your experimental design and the presented data, it would be advisable to revise the manuscript title to more accurately reflect the scope and nature of your research.
- According to CCAC guidelines (2005), in experimental conditions, fish should not be overfed unless they are managed with ad libitum feeding, which is not aligned with the aim of your study. When ad libitum feeding is employed, fish must be carefully monitored, and any excess feed should be promptly removed to prevent water quality deterioration and the proliferation of potentially harmful bacteria and fungi. To address potential concerns regarding the validity of your results and conclusions, it would be beneficial to include additional discussion justifying the rationale behind the use of ad libitum feeding regime as:
(1) Fundulus heteroclitus typically does not experience significant food scarcity in its natural habitat due to its opportunistic omnivorous feeding strategy, which allows it to adapt to variable food availability.
(2) Controlled laboratory conditions, despite numerous uncontrolled factors over the 62-day experiment, along with a consistent diet across all tested groups, provide relevant insights into lipid metabolism processes under specific salinity.
- It is standard practice to fast fish before sampling, similar to procedures used in human biochemistry studies, where 8 hours of fasting is common. However, I am uncertain whether a 24-hour acute fasting period prior to sampling accurately reflects the basal physiological and metabolic state of the fish. While this duration is sufficient for assessing overall fatty acid composition, it could influence parameters such as cortisol, lipase activity, glucose, and, to a lesser extent, total protein, cholesterol, triglycerides, and lactate. This is especially relevant when studying overfed fish. Additionally, a 24-hour fasting period could affect the transcription of genes involved in key PUFA biosynthesis enzymes, as PUFA synthesis is energetically costly, and organisms tend to prioritize essential metabolic pathways during nutritional stress. Although it is understood that your data are derived from a single experimental group, it remains important to explicitly acknowledge this limitation, at least in the discussion and conclusions sections.
While the authors’ response to comments regarding the relationship between salinity stress and molecular responses in the fish brain and eyes was detailed and structured, this information was not included in the manuscript’s introduction. Today, scientific articles are often read by specialists not only within their specific area but also by boundary experts. Therefore, the introduction should clearly emphasize the importance and relevance of the study, highlighting the knowledge gap it addresses.
Dear authors, thank you for your detailed response regarding the salinity concentrations used in your study. However, I still have some points requiring clarification. You mention that salinity exceeds 58 ppt in the “Esteros” of Cádiz Bay. Does Fundulus heteroclitus naturally inhabit these ponds? If so, it would be feasible to source the fish from natural waters to your research: both freshwater and saline ponds with 40 and 60 ppt, and additionally from estuaries with variable salinity levels. Another important aspect is the feeding regime: in natural conditions, fish feed according to their ecological behavior, whereas in laboratory conditions, they are fed ad libitum. Just as a comment, I suggest that, given your study focuses on the physiological responses of fish to a specific salinity condition maintaining (with a fixed 62-day exposure), it would be more appropriate to source fish from natural environments. This approach could enhance the ecological relevance and accuracy of your assessment of lipid metabolism and adaptation mechanisms.
It would be logical to include the feeding details in the Materials and Methods section (from section 3.1), alongside the fish housing information. Furthermore, details about the size of the feed pellets used in the experiment should be provided.
Lines 437–441: Do you believe that hepatosomatic and viscerosomatic indices characterize “liver and gut, organs” in terms of osmoregulation, metabolic, and physiological functions? While it is understandable that authors have preferences for which parameters are most relevant, but these should be justified realistically and supported by scientific evidence. Additionally, the manuscript lacks information about testes condition, stage, or mass of adult fish (additionally to GSI), a factor that can significantly impact fish physiology and should be included.
Lines 442–445: The observed gluconeogenesis could be a consequence of the 24-hour fasting period. This process is a typical metabolic response to short-term food deprivation and should be interpreted accordingly.
Line 450: In the context of “fish osmotic acclimation in response to environmental salinity changes,” it is indeed that fish capable of tolerating such fluctuations efficiently utilize their energetic resources. Furthermore, species adapted to salinity gradients can quickly adjust to salinity fluctuations and utilize their energetic resources effectively. Your results seem to reflect the physiological adjustments (over time) under these specific conditions.
Line 453–456: Given that blood osmolality tends to be lower in freshwater conditions compared to saline and hypersaline environments, how do you explain the high osmolality observed in fish housed in 2 ppt? Please clarify this apparent discrepancy.
Line 545: What functions of cognitive systems in Fundulus heteroclitus are you referring to in this context?
Author Response
I appreciate the effort the authors have invested in revising the manuscript. However, the current version contains areas where clarity, detail, and rationale need further strengthening to meet the journal’s standards of scientific rigor and clarity. The mean and its error alone are insufficient as statistical parameters. Minimum and maximum values are critical as they provide essential insights into the range and distribution of the dataset, helping to identify any outliers. Please provide all the necessary data.
Authors: Thank you very much for your comments. Regarding maximum and minimum values, and as indicated in the previous round of comments, all of them shown in the tables and figures are presented as mean ± SEM. This is a standardized, objective, and widely accepted format in scientific publications, including those in MDPI journals such as Animals. While we appreciate your comments, we believe that the current presentation format already allows readers to interpret the variability of the measurements clearly and intuitively. Moreover, including minimum and maximum values in the tables would require modifying both their format and length, which, in our opinion, would unnecessarily complicate their structure and presentation. In addition to that, the appreciation referred to by this Reviewer was also solved in the Statistical Section of the manuscript, where just in the previous version of our manuscript that “Outliers were identified by ROUT method at Q = 1 %”. For all of these reasons, and mainly to avoid hard interpretation with unnecessary data, we decided to decline this suggestion.
The shallow place of your study is feeding regimes and fasting protocols. The authors should justify their feeding and fasting protocols in relation to the specific objectives of their study, rather than solely referencing widely accepted practices in fish physiology literature.
Authors: We are very sorry if the meaning of this description and further explanation was not understood by this Reviewer. As this practise is commonly carried out in this type of physiological experiments, and in our opinion otherwise would be an incorrect protocol for further metabolic determinations, the following sentence has been included in the new version of our manuscript in case other readers could have the same doubt: “…final sampling to reduce the short-term effects not only induced by recent feeding but also by the differences observed during feeding regarding to feed intake among the 4 experimental salinities/groups (see below). Then, all…”
- The 62-day exposure period constitutes a long-term experiment, likely allowing for the stabilization of physiological responses related to salinity adaptation. While the term “adaptation” can encompass numerous processes, in the context of your study, the observed responses seem to reflect the overall physiological state of fish inhabiting such salinity conditions, rather than rapid or short-term reactions to salinity changes. Since the experiments were conducted in a laboratory setting, the use of the term “environmental” in the manuscript title may be inappropriate. Based on your experimental design and the presented data, it would be advisable to revise the manuscript title to more accurately reflect the scope and nature of your research.
Authors: Thank you very much for your valuable feedback. As this Reviewer states, this word could be understood or interpreted as an ecological term, but it is also true that although our experimental approach was performed in a laboratory, the salinity waters chosen are the external environment where fish were maintained. Even so, and based on your comments, we have revised the manuscript title to make it more representative of the experiment described. Accordingly, the new proposed title is “Physiological effects of water salinity on metabolism and fatty acid biosynthesis in the model fish Fundulus heteroclitus.”
- According to CCAC guidelines (2005), in experimental conditions, fish should not beoverfed unless they are managed with ad libitum feeding, which is not aligned with the aim of your study. When ad libitum feeding is employed, fish must be carefully monitored, and any excess feed should be promptly removed to prevent water quality deterioration and the proliferation of potentially harmful bacteria and fungi. To address potential concerns regarding the validity of your results and conclusions, it would be beneficial to include additional discussion justifying the rationale behind the use of ad libitum feeding regime as:
(1) Fundulus heteroclitus typically does not experience significant food scarcity in its natural habitat due to its opportunistic omnivorous feeding strategy, which allows it to adapt to variable food availability.
(2) Controlled laboratory conditions, despite numerous uncontrolled factors over the 62-day experiment, along with a consistent diet across all tested groups, provide relevant insights into lipid metabolism processes under specific salinity.
Authors: We must apologize, but the Authors do not fully understand your comments since the sentence that you are referring to (“…in experimental conditions, fish should not be overfed unless they are managed with ad libitum feeding…”) is the exact method that we used… We are fully aware of the numerous factors involved in a natural environment and the difficulty of reproducing them under controlled laboratory conditions. Therefore, as you correctly pointed out, in this trial we opted for ad libitum feeding to avoid issues related to over- or underfeeding, deterioration of water quality, and to assess the effects of salinity on food intake, as previously described in our previous rebuttal. Although this approach entails additional technical work, it would not have been feasible using programmable automatic feeders. Nevertheless, in line with your suggestions, we have clarified the feeding protocol in detail in section 2.1 of the manuscript.
- It is standard practice to fast fish before sampling, similar to procedures used in human biochemistry studies, where 8 hours of fasting is common. However, I am uncertain whether a 24-hour acute fasting period prior to sampling accurately reflects the basal physiological and metabolic state of the fish. While this duration is sufficient for assessing overall fatty acid composition, it could influence parameters such as cortisol, lipase activity, glucose, and, to a lesser extent, total protein, cholesterol, triglycerides, and lactate. This is especially relevant when studying overfed fish. Additionally, a 24-hour fasting period could affect the transcription of genes involved in key PUFA biosynthesis enzymes, as PUFA synthesis is energetically costly, and organisms tend to prioritize essential metabolic pathways during nutritional stress. Although it is understood that your data are derived from a single experimental group, it remains important to explicitly acknowledge this limitation, at least in the discussion and conclusions sections.
Authors: First of all, we do not understand the reviewer's appreciation regarding “overfed fish”. As noted in the first round of comments, we agree that fasting can elicit significant metabolic adjustments. The 24-hour fasting period applied before sampling is a standard procedure in nutritional trials with aquatic organisms. This approach was deliberately adopted to reduce variability associated with digestive and postprandial metabolic processes, which could otherwise confound or mask the detection of salinity-induced effects on lipid metabolism and other biochemical parameters. In any case, as stated sometimes before, fish were not overfed, and all experimental groups were subjected to the same fasting period; therefore, any potential effects derived from fasting should have been homogenized across the four groups. Nevertheless, in accordance with your recommendation, this consideration has now been explicitly emphasized in the manuscript.
While the authors’ response to comments regarding the relationship between salinity stress and molecular responses in the fish brain and eyes was detailed and structured, this information was not included in the manuscript’s introduction. Today, scientific articles are often read by specialists not only within their specific area but also by boundary experts. Therefore, the introduction should clearly emphasize the importance and relevance of the study, highlighting the knowledge gap it addresses.
Authors: Following your suggestions, we have proceeded to highlight this information in the manuscript’s introduction.
“At the cellular level, these compounds are present either in free form or as structural constituents of complex lipids such as glycerophospholipids and sphingolipids. In particular, EPA and DHA, as well as LC-PUFA incorporated into sphingomyelin and phosphatidylcholine, are critical for brain and eye integrity and function [25-28]. Accordingly, modulating LC-PUFA biosynthesis in these tissues through environmental factors such as salinity may be critical”.
Dear authors, thank you for your detailed response regarding the salinity concentrations used in your study. However, I still have some points requiring clarification. You mention that salinity exceeds 58 ppt in the “Esteros” of Cádiz Bay. Does Fundulus heteroclitus naturally inhabit these ponds? If so, it would be feasible to source the fish from natural waters to your research: both freshwater and saline ponds with 40 and 60 ppt, and additionally from estuaries with variable salinity levels. Another important aspect is the feeding regime: in natural conditions, fish feed according to their ecological behavior, whereas in laboratory conditions, they are fed ad libitum. Just as a comment, I suggest that, given your study focuses on the physiological responses of fish to a specific salinity condition maintaining (with a fixed 62-day exposure), it would be more appropriate to source fish from natural environments. This approach could enhance the ecological relevance and accuracy of your assessment of lipid metabolism and adaptation mechanisms.
Authors: As noted in the previous round of comments, salinity levels in the "esteros" of the Bay of Cádiz, where F. heteroclitus is widely present, fluctuate considerably depending on seasonality, rainfall regime, tidal dynamics, and human and commercial activities. While some sea-ponds and lagoons remain abandoned, others are actively exploited for sea salt extraction or extensive aquaculture. All these factors substantially affect the temporal continuity not only of salinity conditions but also of other environmental parameters such as water temperature, water column height, turbidity, presence of predators, abundance and nutritional quality of prey, and light availability. Based on this, we believe that maintaining the animals under homogeneous laboratory conditions was the most appropriate approach, as acclimation to these new conditions of stocking are needed and mandatory for this and others types of physiological trials. Nevertheless, for future assays, it would indeed be of interest to include animals originating from the “natural” environment, albeit acknowledging that the "esteros" represent an anthropogenically influenced system.
It would be logical to include the feeding details in the Materials and Methods section (from section 3.1), alongside the fish housing information. Furthermore, details about the size of the feed pellets used in the experiment should be provided.
Authors: As stated before in several comments, and following your instructions, we have expanded the information regarding the feeding protocol in section 2.1. Fish maintenance and experimental design. “Animals were manually fed with commercial feed (Skretting: 57 % protein, 15 % lipid, and 10.5 % minerals. Pellet size: 0.8-1.2 mm) twice a day (10:00 and 15:00 h) until apparent visual satiation (ad libitum); this feeding protocol was chosen to assess the putative effects of salinity on food intake. To prevent water quality deterioration and the potential proliferation of undesirable microorganisms resulting from pellet degradation, any uneaten food was carefully removed after each feeding.”
Lines 437–441: Do you believe that hepatosomatic and viscerosomatic indices characterize “liver and gut, organs” in terms of osmoregulation, metabolic, and physiological functions? While it is understandable that authors have preferences for which parameters are most relevant, but these should be justified realistically and supported by scientific evidence. Additionally, the manuscript lacks information about testes condition, stage, or mass of adult fish (additionally to GSI), a factor that can significantly impact fish physiology and should be included.
Authors: These are classical indices commonly evaluated in aquaculture studies to provide additional information on the physiological status of organs, particularly relevant for proper nutrient absorption and the mobilization of energy resources such as carbohydrates and lipids. While considered alone, they may not provide significant insight; when combined with the biochemical parameters analysed in the assay, they offer a more comprehensive view of the physiological and metabolic response to salinity stress in teleosts [1-4].
Regarding the gonadal stage of the specimens, although we are not specialists in this area, only a minimal gonadal mass was observed, indicating that the individuals were likely not yet sexually mature, either due to age, since they usually reach sexual maturity in the second year [5,6], or due to environmental influences [7]. To clarify this, we have added a note in the Materials and Methods section specifying that the sampled specimens were sexually immature. For future studies, it would be valuable to quantify plasma concentrations of testosterone in males and estradiol-17β in females (if used) to establish a better correlation with gonadal development. Finally, we really do not understand the comment about “or mass of adult fish”, since this information was provided as “Mf: Final Body Mass” just in the first version of our manuscript.
[1] Chen, J.; Cai, B.; Tian, C.; Jiang, D.; Shi, H.; Huang, Y.; Zhu, C.; Li, G.; Deng, S. RNA sequencing (RNA-Seq) analysis reveals liver lipid metabolism divergent adaptive response to low-and high-salinity stress in spotted scat (Scatophagus argus). Animals 2023, 13(9), 1503.
[2] Xiong, Y.; Dong, S.; Huang, M.; Li, Y.; Wang, X.; Wang, F.; Ma, S.; Zhou, Y. Growth, osmoregulatory response, adenine nucleotide contents, and liver transcriptome analysis of steelhead trout (Oncorhynchus mykiss) under different salinity acclimation methods. Aquaculture 2020, 520, 734937.
[3] Zhang, X.; Wen, H.; Wang, H.; Ren, Y.; Zhao, J.; Li, Y. RNA-Seq analysis of salinity stress–responsive transcriptome in the liver of spotted sea bass (Lateolabrax maculatus). PLoS One 2017, 12, e0173238.
[4] Chang, J.C.H.; Wu, S.M.; Tseng, Y.C.; Lee, Y.C.; Baba, O.; Hwang, P.P. Regulation of glycogen metabolism in gills and liver of the euryhaline tilapia (Oreochromis mossambicus) during acclimation to seawater. Journal of Experimental Biology 2007, 210(19), 3494-3504.
[5] Kneib, R.T.; Stiven, A.E. Growth, reproduction, and feeding of Fundulus heteroclitus (L.) on a North Carolina salt marsh. Journal of experimental marine biology and ecology 1978, 31(2), 121-140.
[6] Abraham B.J. Species profiles: life histories and environmental requirements of coastal fishes and invertebrates (mid-Atlantic)–mummichog and striped killifish. U.S. Fish and Wildlife Service Biological Report 1985, 82(11-40).
[7] Shimizu, A. Effect of photoperiod and temperature on gonadal activity and plasma steroid levels in a reared strain of the mummichog (Fundulus heteroclitus) during different phases of its annual reproductive cycle. General and comparative endocrinology 2003, 131(3), 310-324.
Lines 442–445: The observed gluconeogenesis could be a consequence of the 24-hour fasting period. This process is a typical metabolic response to short-term food deprivation and should be interpreted accordingly.
Authors: As indicated, the potential influence of fasting on metabolism has been highlighted in the Discussion, and in any case, as all fish were subjected to the same protocol, differences observed are produced by the environmental factor studied: salinity.
Line 450: In the context of “fish osmotic acclimation in response to environmental salinity changes,” it is indeed that fish capable of tolerating such fluctuations efficiently utilize their energetic resources. Furthermore, species adapted to salinity gradients can quickly adjust to salinity fluctuations and utilize their energetic resources effectively. Your results seem to reflect the physiological adjustments (over time) under these specific conditions.
Authors: Thanks for your appreciation.
Line 453–456: Given that blood osmolality tends to be lower in freshwater conditions compared to saline and hypersaline environments, how do you explain the high osmolality observed in fish housed in 2 ppt? Please clarify this apparent discrepancy.
Authors: This Reviewer is right, and our results are unusual regarding the hypo-osmotic environment, since it is known that under this condition an increase in this parameter can be observed after an acute osmotic challenge as a consequence of an acute stress caused and the endocrine activation of the HPI axis with the final cortisol release, but not after a chronic exposure to low salinity waters. This hormone assists with the activation of the hyperosmotic regulation, but curiously, in our case, cortisol is significantly lower in the 2 ppt group. For this reason, during our analyses, we decided to measure all samples from these experimental groups again, together with several samples of the other Groups, to be sure that no methodological or technical errors were introduced, being all results as presented herein. Thus, although we previously assumed this deviation in accordance with the typical U-shape response caused in important transporters that supports this osmotic regulation, we really thank and appreciate the comment performed by this Reviewer, and we have introduced this idea in the new version of our manuscript, highlighting that further investigation would be necessary to fully understand this intriguing results.
Line 545: What functions of cognitive systems in Fundulus heteroclitus are you referring to in this context?
Authors: The cognitive system of fish is complex and multifaceted, as they possess a wide range of cognitive abilities that enable them to adapt to novel situations and avoid potential threats. However, since the cognitive capacities of the fish used in this trial were not assessed, we consider it more appropriate to replace the term “cognitive system” with “neural system”, thereby referring specifically to the brain and visual system.
